# A Study of the Simulation and Analysis of the Flow Field of Natural Convection for a Container House

**Hsin-Hung Lin [1,2,]\* and Jui-Hung Cheng [3]**

[1]  Department of Creative Product Design, Asia University, Taichung City 41354, Taiwan
[2]  Department of Medical Research, China Medical University Hospital, China Medical University, Taichung City 406040, Taiwan
[3]  Department of Mold and Die Engineering, National Kaohsiung University of Science and Technology, Kaohsiung 80778, Taiwan; rick.cheng@nkust.edu.tw
[\*]  Correspondence: hhlin@asia.edu.tw or a123lin0@gmail.com; Tel.: +886-04-2332-3456-1051

**Abstract:** Natural disasters, such as earthquakes, windstorms, and tsunamis, can occur all over the world, and disasters caused by human factors, such as civil wars, are also a source of major disturbance. The temporary rehousing of the population is a major problem when disasters occur. The installation of the combination house is time consuming, and tents cannot be used in the event of strong rain and wind; therefore, the container house is the most effective way of solving the rehousing problem. Natural ventilation is the main factor affecting the indoor air quality, thermal comfort, and health inside a container house, and solar radiation heat can also affect temperature changes inside. The air flow field inside a dwelling is very complex, and its flow mode is affected by inlet wind speed, inlet temperature, solar radiation heat, and the size of doors and windows, etc. In this paper, the influence of natural ventilation on the ventilation inside container houses is analyzed. Assuming that there is complex fluid motion in the activity space of the container house, it is not easy to use conventional methods to predict the flow rate. Based on the correlation analysis motion between the corresponding internal flow rates, the calculation and application method of flow is simplified from the results of the wind speed coefficient obtained previously. In addition, an analysis of flow characteristics in the container house is made; simulation analysis in the container house is made by carrying out the numerical analysis of several factors, including velocity field and temperature field. The variation state of the temperature of the environment and a numerical variation of the three-dimensional space are obtained by numerical calculation; the standard $k$-$\varepsilon$ turbulence model is adopted to describe the turbulence phenomena of the fluid, and the mathematical model matched by B-spline surface is used for data analysis through the surface algorithm in order to deal with complex simulation data. The research results show that, regarding the influence of natural ventilation on container houses, the ideal relative position of openings includes the combination of asymmetric windows, followed by the central positioning of the door. The four-opening configurations, where better natural ventilation performance can be achieved, are located at different diagonal positions. The average flow velocity vector form, velocity amplitude, radiation temperature distribution, and the effect of the air volume coefficient of temperature change are analyzed. The research results show that the design of container houses can meet the requirements of air flow, such as the energy consumed by the thermal comfort space. Measurements taken over time and algorithms can also check the residents' indoor natural ventilation and provide health care by the use of various sensors.

**Keywords:** natural ventilation; openings configuration; CFD; solar radiation; cross ventilation

## 1. Introduction

Natural disasters, such as the recent earthquake in Japan, often occur due to environmental changes. When a disaster occurs, one important consideration is the rehousing of survivors. In recent years, due to many major natural disasters, such as earthquakes, wind disasters, tsunamis, and so on, a large amount of temporary accommodation has been required. Container houses (as shown in Figure 1) can be used for temporary rehousing.

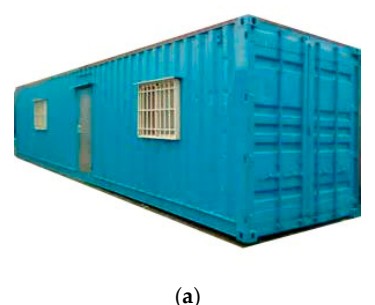　　　　　　　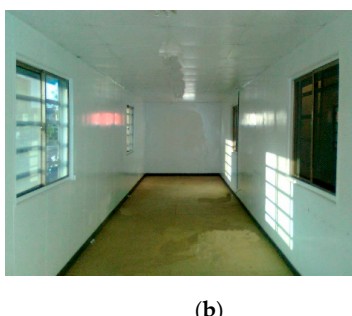

(**a**)　　　　　　　　　　　　　　　　　　　　　　　(**b**)

**Figure 1.** Container House (**a**) Container house appearance, (**b**) Container house interior.

In the indoor environment, natural ventilation is often used as a low-energy environmental solution to improve indoor thermal comfort. In hot and humid tropical climates, natural ventilation is particularly effective because it maintains a balanced indoor relative temperature and prevents indoor stuffiness. Natural ventilation can be used as an effective way to significantly reduce the energy use of air cooling systems in container houses by ventilation through open doors and windows. This also provides natural ventilation and fresh air. Natural ventilation and heat dissipation design elements are mostly ignored in modern planning and architectural design. Doors and windows can conveniently provide the function of natural ventilation for heat dissipation, and ventilation is a natural mechanism in the process of air exchange between indoor and outdoor spaces. In this article, many aspects of ventilation, including turbulence effect, is analyzed, and simulation values are used to simulate technology applications. According to the comparison in flow between numerical simulation and measurement, the validity of the simplified estimation method is also verified.

### 1.1. Natural Ventilation, CFD (Computational Fluid Dynamics)

An average value was used to measure the result of the wind tunnel test LES (large eddy simulation) based on internal velocity [1], the surrounding building model, and pressure distribution. Three ventilation cases of the model were verified; namely, single and double-sided ventilation and windward wall openings, leeward walls of openings for single-sided ventilation, and cross ventilation. Research was conducted regarding these aspects. The CFD model is very sensitive to how boundary conditions are set, and a simulated indoor and outdoor environment is needed to construct the model. With efforts being made in this regard, the error value in the calculation of ventilation rate and simulation analysis is less than 10%. The difference survey shows that the effect of natural ventilation may be strengthened or interacted, and the wind speed analysis method, through a series of spatial velocity patterns, is verified [2].

The linear regression analysis method is also verified. The ideal mixed ventilation, with an average level of 56% being saved, is selected using the experiment method and through a case explanation [3]. Based on the related characteristics between corresponding internal pressure and flow velocity, the estimation method of flow is simplified [4]. Based on the analysis of empirical data, a ventilation tower with a domed roof is able to generate ACH (air change per hour, ACH) with extraction air flow of 10,000 cubic m/r at an external wind speed of 0.1 m/s. If the experimental house has no wind speed or natural ventilation tower, only cross-ventilation can be relied on, and the change of air is only 7 ACH [5]. In terms of mass flow linearity of the speed change between window and indoor

air, if the wind speed changes linearly, the distribution of the pressure coefficient indicates that the ventilation inlet is of an appropriate type [6]. It is verified that energy saving has the same performance to provide sufficient airflow velocity and acceptable thermal comfort, cross-reference is needed, so that the performance of each ventilation strategy can become another solution, that is, ventilation aspects can be selected or arranged in the design [7]. It is found from the result that the comfort time in summer is expanded from 37.5% indoors to 56.3% in the laboratory. For a typical 5-kilowatt wall-mounted split air conditioner, this extension can save about 2700 kilowatt-hours of electricity per residential unit per hour over four months in summer [8].

The following conclusions are drawn from the research: (1) The ventilation rate is adjusted according to the pressure difference along the opening height above the experimental confirmation model and through the mean value and fluctuation calculated by opening. The pulsating flow of ventilation rate fluctuation and vortex penetration are a combination. This survey adopts full spectrum analysis to find vortex penetration effect, which is proven to be a major factor when the wind is parallel and full open [9]. (2) The CFD simulation of single-sided natural ventilation confirms that the differences between model prediction and CFD and experimental data all are less than 25%. (3) It is found from this study that wind incidence angle is 0° when vortex penetration rate is zero because of zero parallel velocity, the angle of low penetration rate is about 70°, due to the lower absolute pressure coefficient. The ventilation rate can make the increase of the opening with non-linear opening size being raised to the ground decrease. It is found that the natural ventilation performance of two different parameters (window position and building facing different directions) can have a positive effect, but does not change the yield improvement of all input parameters, because the effect of backpressure changes in the position of the door [10]. It is found that further experiments and numerical analyses in the study show that the essential characteristics of air flow in the passage between buildings connecting the basement can provide a function of sustainable development in ventilation design of the adjacent basement, therefore, healthy environment is suitable for living. Solar chimneys are a good passive ventilation strategy used to enhance natural ventilation and thermal comfort for occupants [11]. While some studies have focused primarily on understanding heat transfer processes and fluid mechanics inside ventilation chimneys, others have focused on flow performance and thermal conditions within the auxiliary building [12]. In order to calculate the distribution ratio in thermal flux and heat, and for air velocity and heat transfer rate in the design of such natural ventilation system [13]. CFD and standard $k-\varepsilon$ turbulence model are adopted. The model validation calculus shows the calculated results and experimental measurements in the existing literature [14]. Heat distribution, proportion and increase of two air flow rates and heat transfer coefficients are in a wide range, while cavity size and total heat input, in general, are more evenly heat-distributed at the vertical inlet and outlet of the two space walls, with larger flow rate, but smaller heat transfer coefficient [15]. Replacement of mechanical ventilation and natural ventilation can reduce energy consumption by nearly 60% in summer. More energy can be saved, which is achieved by automatic correlation and automatic control method between intake and exhaust openings from one side and natural and mechanical ventilation systems from the other side [16]. Through theoretical analysis and experimental research. Mixed ventilation with vertical temperature distribution is predicted according to the theory of block model and the theory of natural ventilation [17]. This generalized vertical temperature distribution is drawn from the theoretical model of ventilation prediction of mixed atrium buildings. Suggested fresh air is provided under relatively low external wind speed. These results indicate that the ventilation equipment system is a viable alternative ventilation even in urbanized and shaded applications [18]. The main purpose of the study is to establish a guide line to improve natural ventilation in the daytime [19]. A double-skin curtain wall is built in the office, which determines the size of the necessary opening of the double windows to be opened. The experiment confirms the rate of ventilation 4 ACH in each office and various wind speed conditions. The study results show that natural ventilation increases the window-wall ratio by 0.24, which greatly improves the indoor thermal conditions and the four horizontal sunshade devices needed to further improve the indoor thermal comfort [20]. The study results show that in passive and potential

use of the natural ventilation cooling system, the direction of air flow in the building in summer can significantly improve the way of natural ventilation. According to the size ratios of 1:11:1.44 and 1:1.7 of three different geometric simulated buildings and the wind speed of 1–1.5 m/s, both the square and the two rectangular buildings meet the requirements of indoor thermal comfort [21].

*1.2. Solar Radiation*

Calculated out surface solar radiation throughout Myanmar, and monthly average daily solar radiation. Then the results are shown as solar resource mapping. Myanmar's solar radiation is mainly affected by topography and tropical monsoon climate [22]. With long-term annual average solar radiation, it is found that it is 18.3 MJ/m on average throughout the regions of Myanmar. The amount of radiation from the direct beam at the adjacent station is collected from the main monitoring station, global level measurements and adjacent solar radiation station between 12:00 to 14:59 on 1 September 2010 [23]. The calculated value of the direct radiation is then compared with the actually observed value.The radiation correlation coefficient (r), maximum correlation coefficient (r) (0.96–0.99) and determination coefficient (R2) (0.92–0.98) of the global solar radiation per hour at different locations and global solar radiation value per hour are predicted through the calculation for quantitative evaluation [24]. The research of solar ultraviolet radiation time and spectra depends on changes in a solar activity cycle [25]. Solar radiation was estimated in northern Malaysia in 2006. The relationship between the average air temperature and the average sun radiation is linear, and the value of R2 is 0.5593, which is quite good, and the change in the average temperature shows the average solar radiation [26]. In terms of solar radiation data collected in Belgium, two surface incoming global short-wave radiation products from different satellite application equipment (LSA-SAF and retrieval CM-SAF) are assessed in accuracy for the first time [27]. put forward the collection and processing of solar radiation and other meteorological data in the process of the actual measurement of solar direct radiation in Abu Dhabi (24.43 N, 54.45 E) under the existing meteorological conditions [28]. The research results show that: (1) The radiation differences based on solar energy and based on direct solar radiation have no statistical significance [29], and there is a strong linear correlation based on total solar radiation; (2) The difference of average daily solar radiation of 12 in the growing season varies from 0.08 to 13.28 MJ m$^2$. (Adel) [30] The results obtained indicate a very high potential for solar radiation, and an analysis of the slope in all Oman throughout the year is made. (Hüsamettin Bulut 2004) [31] In the study, year-round daily solar radiation in seven provinces in southeastern Anatolia Turkey is referred to in the tests by using at least 14 years of measurement data [32–34]. Different situations can be reviewed by adjusting the fan speed in the simulation model, and the airflow characteristics can be determined by rotating the simulation model to evaluate thermal comfort characteristics. When the ceiling fan circulates in the area, the thermal comfort is significantly enhanced. By maintaining a reasonable level of thermal comfort, higher room temperature or higher thermal load can be allowed, so that a sustainable environment can be maintained without affecting indoor thermal comfort or energy efficiency [35–37].

## 2. Research Model

Natural ventilation is the simulation method for the verification of the container house, and the model for establishing the actual simulation shown in Figure 2 [1,5] is the research model. And the interior design of the container house is the design modeling of square windows and doors. The container house is equipped with air inlet and outlet with eight different models, which requires very long calculation time to simulate the authenticity. Therefore, there is a challenge, namely minimizing computation time while maximizing accuracy [6]. The establishment of the model is based on the natural ventilation of the container house. In order to simulate the flow field behavior close to the actual space, the sizes of the simulated actual model established are 12.19 m × 2.44 m × 2.59 m (length × width × height), including the wall thickness of 0.1 m, as shown in Figure 2. As for the thickness on the surface (wall) and under the ground, it does not need to be considered, because it does

not have any influence on the flow field. The interior of the container house is designed with a square window with the size of 67 cm × 67 cm (length × height), and the size of the door is 120 cm × 210 cm (length × height), as shown in Table 1. In this study, the effect of natural ventilation openings on the indoor wind field and radiation heat of the container house, and the air outlet and inlet taking the window and the door of the container house as ventilation parts are taken as the setting of the openings. Therefore, five parts are selected for the combination configuration of opening positions, including two inlet windows, two outlet windows and the door as the ventilation opening which draws the wind into the house. Through the combination of various opening positions, the simulation is made to compare the changes of different opening positions to the internal temperature of the container house and to evaluate the advantages and disadvantages of natural ventilation convection. The selected analysis space has eight different changes. The opening positions of the container house are as shown in Figure 3 and Table 2. A total of 8 ventilation opening modules are designed for numerical simulation [21].

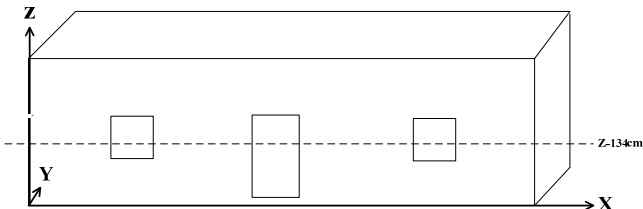

**Figure 2.** Research model.

**Table 1.** Size table of the container house, window and door.

| | |
|---|---|
| Container house size | 12.19 m long/2.44 m wide/2.59 m high |
| window | 67 cm × 67 cm |
| Door | 120 cm × 210 cm |

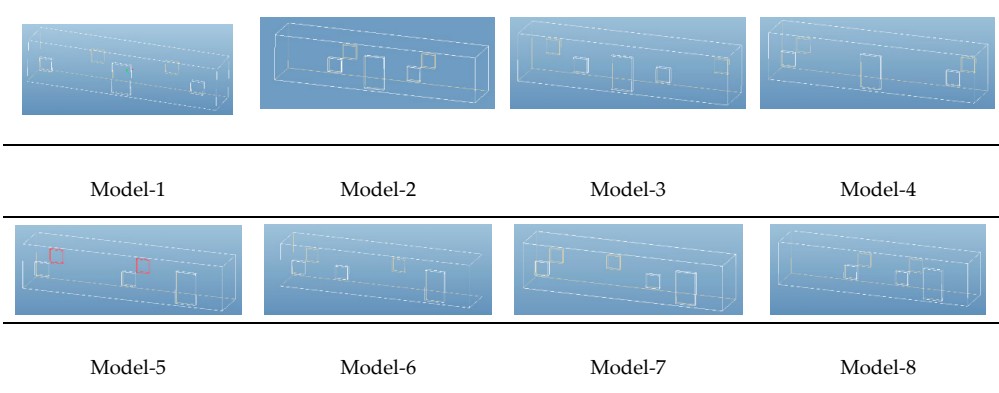

| Model-1 | Model-2 | Model-3 | Model-4 |
|---|---|---|---|

| Model-5 | Model-6 | Model-7 | Model-8 |
|---|---|---|---|

**Figure 3.** Research model styles.

**Table 2.** Size table of relative positions of window and door.

| cm | Window 1 | Window 2 | Window 3 | Window 4 | Door |
|---|---|---|---|---|---|
| Model-1 | X80/Y0/Z95.5 | X1060/Y0/Z95.5 | X330/Y235/Z95.5 | X810/Y235/Z95.5 | X550/Y0/Z20 |
| Model-2 | X330/Y0/Z95.5 | X810/Y0/Z95.5 | X330/Y235/Z95.5 | X810/Y235/Z95.5 | X550/Y0/Z20 |
| Model-3 | X330/Y0/Z95.5 | X810/Y0/Z95.5 | X80/Y235/Z95.5 | X1060/Y235/Z95.5 | X550/Y0/Z20 |
| Model-4 | X80/Y0/Z95.5 | X1060/Y0/Z95.5 | X80/Y235/Z95.5 | X1060/Y235/Z95. | X550/Y0/Z20 |
| Model-5 | X80/Y0/Z95.5 | X610/Y0/Z95.5 | X80/Y235/Z95.5 | X610/Y235/Z95.5 | X940/Y0/Z20 |
| Model-6 | X80/Y0/Z95.5 | X360/Y0/Z95.5 | X80/Y235/Z95.5 | X360/Y235/Z95.5 | X940/Y0/Z20 |
| Model-7 | X80/Y0/Z95.5 | X760/Y0/Z95.5 | X80/Y235/Z95.5 | X760/Y235/Z95.5 | X940/Y0/Z20 |
| Model-8 | X430/Y0/Z95.5 | X760/Y0/Z95.5 | X430/Y235/Z95.5 | X760/Y235/Z95.5 | X940/Y0/Z20 |

## 3. Research Methods

The air flow in the container room is a case of turbulence. The development of fluid mechanics is still unable to master the phenomenon of turbulent flow completely. It must be simulated by mathematical model. In this paper, one of the widely accepted modes—standard mode of turbulent flow is used to correlate the flow and turbulent conditions. Apart from velocity and temperature, it must first be emphasized that turbulent flow is the most common but also the most complex way of fluid flow.

Assume that the calculated flow: The flow is stable, three-dimensional, viscous, turbulent, incompressible, and with appropriate boundary conditions. The standard k-$\varepsilon$ model is considered to be an acceptable accurate simulation of the flow characteristics established by turbulent flow, the numerical value has passed the systemic solution, the steps corresponding to the CFD system are described below.

### 3.1. Governing Equation

Under the 3D Cartesian Coordinate, the governing equation has the form:

1.  Continuity Equation:

$$\frac{\partial u}{\partial x} + \frac{\partial v}{\partial y} + \frac{\partial w}{\partial z} = 0 \tag{1}$$

2.  Momentum Equation:

    *X* direction:

$$\frac{\partial u}{\partial t} + \frac{\partial (u^2)}{\partial x} + \frac{\partial (uv)}{\partial y} + \frac{\partial (uw)}{\partial z} = -\frac{1}{\rho}\frac{\partial P}{\partial x} + v\left[\frac{\partial^2 u}{\partial x^2} + \frac{\partial^2 u}{\partial y^2} + \frac{\partial^2 u}{\partial z^2}\right] \tag{2}$$

    *Y* direction:

$$\frac{\partial v}{\partial t} + \frac{\partial (uv)}{\partial x} + \frac{\partial (v^2)}{\partial y} + \frac{\partial (vw)}{\partial z} = -\frac{1}{\rho}\frac{\partial P}{\partial y} + v\left[\frac{\partial^2 u}{\partial x^2} + \frac{\partial^2 u}{\partial y^2} + \frac{\partial^2 u}{\partial z^2}\right] \tag{3}$$

    *Z* direction:

$$\frac{\partial w}{\partial t} + \frac{\partial (uw)}{\partial x} + \frac{\partial (vw)}{\partial y} + \frac{\partial (w^2)}{\partial z} = -\frac{1}{\rho}\frac{\partial P}{\partial z} + v\left[\frac{\partial^2 w}{\partial x^2} + \frac{\partial^2 w}{\partial y^2} + \frac{\partial^2 w}{\partial z^2}\right] \tag{4}$$

3.  Energy Equation:

$$\frac{\partial T}{\partial t} + \frac{\partial (uT)}{\partial x} + \frac{\partial (vT)}{\partial y} + \frac{\partial (wT)}{\partial z} = \alpha\left(\frac{\partial^2 T}{\partial x^2} + \frac{\partial^2 T}{\partial y^2} + \frac{\partial^2 T}{\partial z^2}\right) + \frac{q}{pC_P} \tag{5}$$

$$\frac{\partial (\rho\phi)}{\partial t} + \frac{\partial (\rho\phi u)}{\partial x} + \frac{\partial (\rho\phi v)}{\partial y} + \frac{\partial (\rho\phi w)}{\partial z} = \frac{\partial}{\partial x}\left(\Gamma\frac{\partial \phi}{\partial x}\right) + \frac{\partial}{\partial y}\left(\Gamma\frac{\partial \phi}{\partial y}\right) + \frac{\partial}{\partial z}\left(\Gamma\frac{\partial \phi}{\partial z}\right) + s \tag{6}$$

Diffusive term, *S* Source term, $\frac{\partial (\rho\phi)}{\partial t}$ unsteady term, when assuming steady state, this term is not considered. Symbol $\phi$ with then *u, v, w, k, $\varepsilon$* dependent variables. Please refer to Table 3; $\Gamma$ is the diffusion system corresponding to the change, *u, v, w* Respectively *x, y, z* Velocity component in direction.

According to the basic principle of the finite volume method, the required calculation space is divided into many small control volumes. After the volume is divided, the equations of the mass, energy, and dynamics of the carcass can be converted into the substitutional equations as follows:

$$\frac{\partial}{\partial t}\int_v (\rho\varphi)dV + \int_A \vec{n}\cdot\left(\rho\varphi\vec{V}\right)dA = \oint_A \vec{n}\cdot\left(\Gamma_\varphi\nabla\varphi\right)dA + \int_V S_\varphi \cdot dV \tag{7}$$

Among them, $\oint_A \vec{n}\cdot(\rho\varphi\vec{V})dA$ Convective term, $\oint_A \vec{n}\cdot(\Gamma_\varphi\nabla\varphi)dA$, Diffusive term, $\oint_V S_\varphi\cdot dV$ generation term.

**Table 3.** Comparison table.

| Equation | $\psi$ |
|---|---|
| Continuity | *1* |
| X-momentum | *u* |
| Y-momentum | *v* |
| Z-momentum | *w* |
| Energy | *I* or *T* |

### 3.2. Turbulent Mode

Turbulent flow causes mutual exchanges of momentum, energy and concentration change between fluid media, and causes a lot of fluctuations, which are of small scale and high frequency. Therefore, when turbulent flow is simulated, the control equation is first processed to filter out turbulent components of very high frequency or very small scale. However, the modified equation may contain variables unknown to us, and the turbulent flow model needs to use known variables to determine these variables. In this study, when turbulent flow is simulated, the standard $k-\varepsilon$ turbulent flow model is selected to calculate the flow field.

### 3.3. Standard $k-\varepsilon$ Disturbance Mode

The standard $k-\varepsilon$ turbulent flow model adopted has become a major tool in turbulent flow field calculation, because of its wide range of application and reasonable accuracy. The standard $k-\varepsilon$ turbulent flow model is a semi-empirical turbulent flow model. The transmission equations of turbulent kinetic energy *(k)* determining turbulent flow transmission and dissipation rate *(ε)* are derived in the following form mainly based on the basic physical control equation:

Turbulent kinetic energy equation (k)

$$\frac{\partial}{\partial t}(\rho k) + \frac{\partial}{\partial x_i}(\rho k u_i) = \frac{\partial}{\partial x_j}\left[\left(\mu + \frac{\mu_t}{\sigma_k}\right)\frac{\partial k}{\partial x_j}\right] + G_k + G_b - \rho\varepsilon - Y_M \tag{8}$$

Dissipating martingale equation (ε)

$$\frac{\partial}{\partial t}(\rho\varepsilon) + \frac{\partial}{\partial x_i}(\rho\varepsilon u_i) = \frac{\partial}{\partial x_j}\left[\left(\mu + \frac{\mu_t}{\sigma_\varepsilon}\right)\frac{\partial\varepsilon}{\partial x_j}\right] + C_{1s}\frac{\varepsilon}{k}(G_k + C_{3\varepsilon}G_b) - C_{2g}\rho\frac{\varepsilon^2}{k} \tag{9}$$

Dysentery $(\mu_t)$

$$\mu_t = \rho C_\mu \frac{k^2}{\varepsilon} \tag{10}$$

In the equation $G_k$ Represents the turbulent kinetic energy generated by the layer speed, $G_b$ Turbulent kinetic energy, $Y_M$ Fluctuations due to excessive diffusion in compressible turbulence, $\sigma_k$ $C_{1\varepsilon}$, $C_{2\varepsilon}$ and $C_{3\varepsilon}$ They are all experiences, and their recommended values are shown in Table 4.

The $k-\varepsilon$ model is an equation assuming that the field is completely turbulent and the molecular viscosity can be ignored. Therefore, the standard $k-\varepsilon$ model has better results for the complete turbulent field calculation.

**Table 4.** Normal system of standard k−ε turbulence model.

| $C_{1\varepsilon}$ | $C_{2\varepsilon}$ | $C_\mu$ | $C_k$ | $C_\varepsilon$ |
|---|---|---|---|---|
| 1.44 | 1.92 | 0.09 | 1.0 | 1.3 |

*3.4. Boundary Conditions*

In this paper, the influence of solar radiation on the container house is discussed under the environmental setting of the boundary conditions. Its main purpose is also to consider the environment around the physical model and the physical phenomenon of the object. The most important thing is to conform to the real physical phenomenon, otherwise the results of the overall simulation calculation will be affected. The boundary of this case study includes: inlet boundary condition, outlet boundary condition and wall boundary condition. As for simulating the thermal environment produced by the actual sun on the container house, the influence of thermal radiation caused by solar rays entering the computational domain is directly considered by using the conditions of longitude, latitude, time zone, date, time and sunshine factor, etc. in the region of the literature, the sunshine factor of which is the main factor to control the amount of solar radiation. [23–25]. Compared with the method of directly giving constant temperature or constant heat flux on the boundary condition setting of the building wall, the solar load model is closer to the actual thermal environment condition. The items are described in Table 5 below:

**Table 5.** Details of simulated boundary conditions and calculation parameters.

| ventilation | Inlet boundary condition | Input = opening width (m), air temperature: 25 °C/30 °C/35 °C/40°C (°C), and inflow velocity:10 m/s |
|---|---|---|
| | Outlet boundary condition | No sliding condition |
| | Outlet | Outlet static pressure is 0 |
| Wall, radiant surface | speed | Generalized record file method, no sliding condition |
| | Solar axis radiation heat | In the literature, the maximum instantaneous radiant energy falling on the surface during the summer heat can be estimated by 1041 W/m$^2$ [22,23,31]. The surface temperature is determined on the solid surface, and thermal equilibrium is simulated through radiation and conduction. The convective heat transfer coefficient is fixed on the radiation surface: 1000.0 [W/(m$^2$ °C)] |
| | Humidity | Radiation surface: AH (absolute humidity) gives the corresponding saturated vapor pressure, the surface temperature of the radiation cooling surface is lower than the dew point temperature of air, water vapor condensation occurs, in other cases, AH = 0. Humidity heat transfer coefficient is calculated. |
| | Surface emissivity | Wall: 0.9, symmetric plane: 0.0 |
| Grid system | | Grid: 560,000 control volumes with minimum control quantity of 1 |

## 4. Model Validation

In this study, for the comparison between the simulation result of the validation method and the experimental result, available application program of reliable data for the ability to perform validation calculus and for CFD accurate prediction of the form of indoor environment, Jiang et al. conducted similar comparative experiments of numerical simulation results [1]. The cubic model of wind tunnel experiment and the natural ventilation of the surrounding flow field are carried out. The wind tunnel has a cross-section of 2 m and a height of 1 m wide. The maximum wind speed in the tunnel is 12.0 m/s, and the time of its change measurement process is within 2%. The cube model is a model with a central part of the vertical line 250 mm × 250 mm × 250 mm where each line

has 18 measurement points to be measured. In this paper, it is revised that the actual model size is 12.19 m long × 2.44 m wide × 2.59 m high, the building in Figure 4 is simulated and compared with the experimental results of Jiang [1] et al. It is found that the simulation results are consistent with the experimental results in the wind speed measurement points in the symmetric part of the model in the building, as shown in Figure 5. Figure 6 Status in the numerical range model of the central part, the central part where the grid distribution lies and construction carried out on the grid plane where the velocity is measured. The velocity is measured along the vertical line 10, and the measurement range from the floor in the simulated pipeline is from 0 to the height of 2H. Figure 5 shows the positions of these measurement lines. Figure 6 Status in the numerical range model of the central part and grid distribution, and Figure 7 shows the experimental velocity components between the simulated velocity components. The value analysis of the velocity components can include the upper central part of the model from the vertical line. The experimental results and detailed numerical simulation are compared. External and internal models of average velocity distribution are analyzed, including the flow mode of wind speed on the model. Figure 6 shows the average velocity U/Uref, and compares the distribution models of 9 upper central vertical lines V/Uref (U and V are average velocity respectively). With the same air intake wind profile at *Y*-axis 3H, it is obvious that it can be known from simulated results that the standard $k-\varepsilon$ model generates matching.

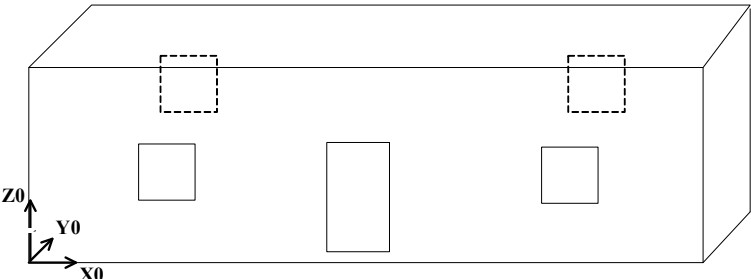

**Figure 4.** Simulation model for validation.

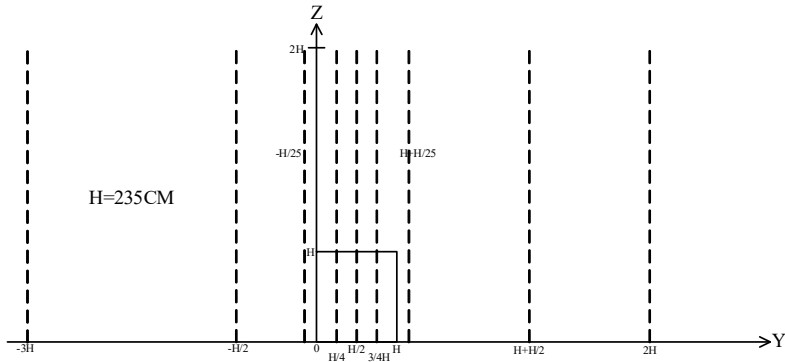

**Figure 5.** Wind velocity measurement points in the symmetric part of the model.

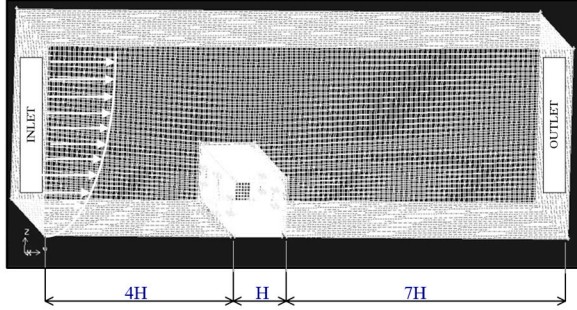

**Figure 6.** Status in the numerical range model of the central part and grid distribution.

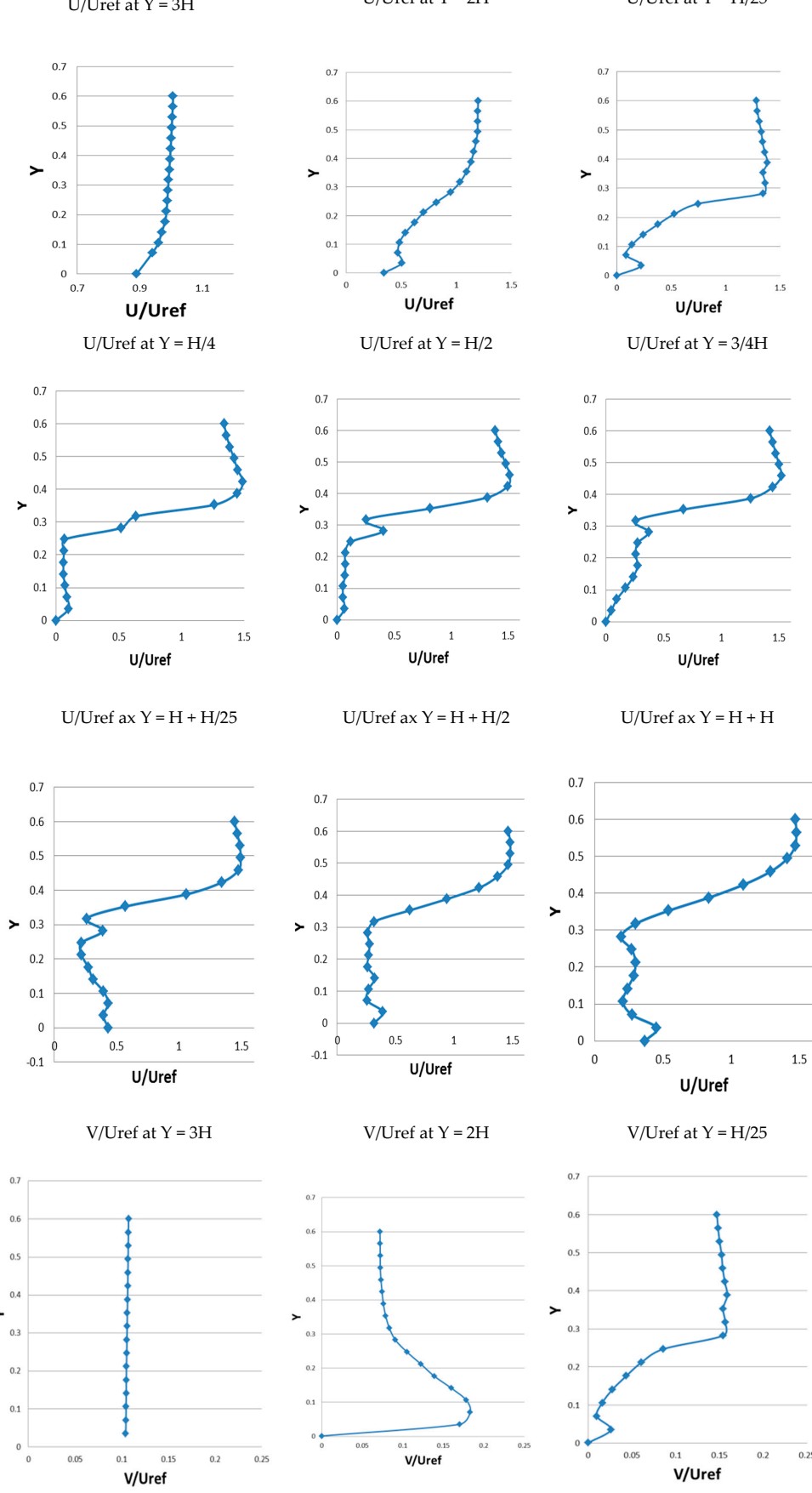

**Figure 7.** *Cont.*

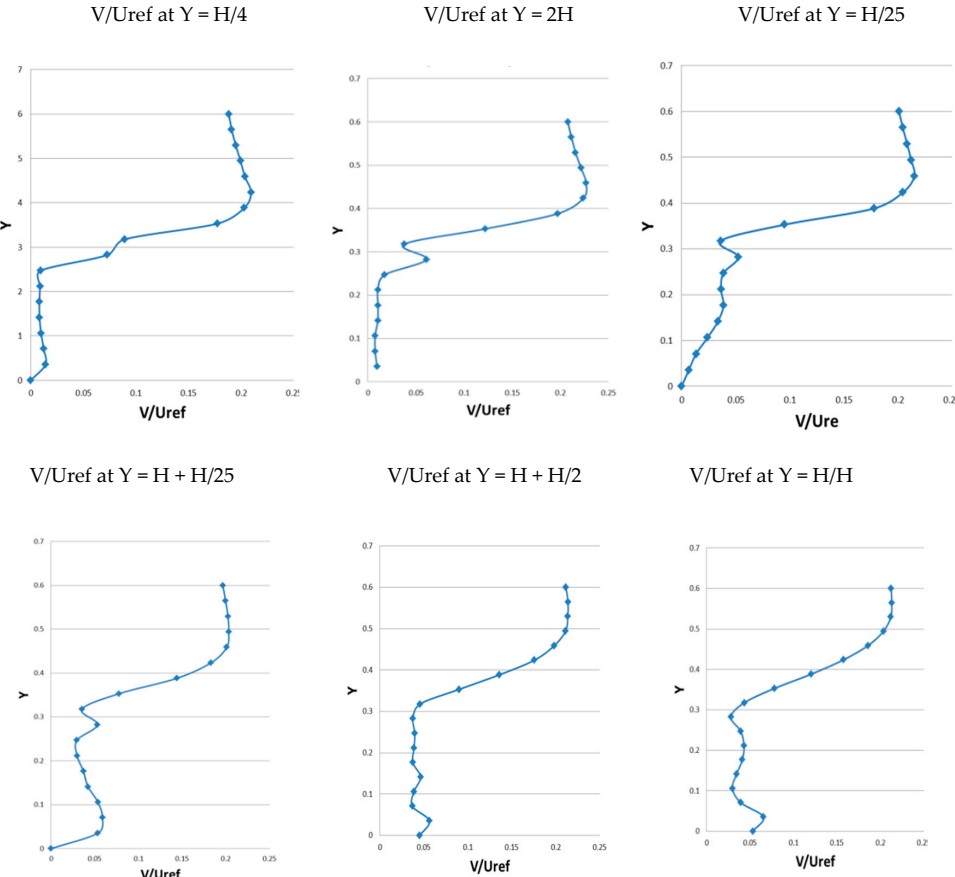

**Figure 7.** Average velocity U/Uref and V/Uref on 9 vertical boundary lines in the center of the model.

## 5. Results and Discussion

This study mainly discusses the indoor effects of natural ventilation on container houses and solar radiation on container houses to ensure the physical comfort of occupants of container houses. In terms of natural ventilation, four windows and one door are set up in eight models to study the ventilation effect, and the influence of natural ventilation flow field on solar radiation container house is analyzed. The flow field ventilation environment under the condition of natural ventilation needs to be predicted for the container house, and the natural ventilation wind speed is set to be 10 m/s. Then, indoor air flow field is analyzed, and solar radiant heat is discussed. In order to make 3D analysis as surface fitting, B-spline is used in this study as surface result analysis of fitting function.

### 5.1. Container House Natural Ventilation Analysis

Figure 8 shows the 3D surface analysis diagram of the flow distribution inside the container house. Under the condition of natural ventilation, air flows out from two windows, and the average velocity distribution along the XY plane is shown in Figure 8 (Velocity, flow and 3D distribution diagram). Obviously, the velocity in the container house is under the condition of high convective ventilation. It is anticipated that in case of outlet ventilation at the stable velocity state of the container house and under natural ventilation, the outlet ventilation is slightly higher than that at the inlet of the container house, wind velocity is too concentrated at air outlets of Model-5 and 8, thus high air volume is generated. While for Model-5–8, the surrounding part at the position of 1100 cm faces the air vortex from the wall. While the windows are under natural ventilation, simultaneous natural ventilation inflow and outflow situations are allowed, therefore, due to the fact that each window is the same, in the steady state, the inflow through windows 1 and 2 and the door should be exactly the same under natural ventilation. However, the small difference in inflow is attributable to the discrete calculation,

which leads to asymmetric truncation error in the computational domain. In terms of flow distribution through the interior of the central container house along the YZ plane, it is observed by observing the relatively high wind velocity zone that in the vertical surface of the upper wind, the velocity difference in the container house is very large, and the flow quantity near the wall of the container house of Model-4 (Figure 8d) is higher than that in the center.

　　　Figure 8a shows that when the uniform distribution of the inflow velocity is 1.92 m/s and when the maximum value of inflow average velocity is 5.12 m/s, the maximum wind velocity value of the surface velocity distribution diagram is at the place of 610 cm, Figure 8b shows that when the uniform distribution of the inflow velocity is 2.05 m/s and when the inflow average velocity is 4.52 m/s, the maximum wind velocity value of the surface velocity distribution diagram is at the place of 610 cm, Figure 8c shows that when the uniform distribution of the inflow velocity is 1.49 m/s and when the inflow average velocity is 4.75 m/s, the maximum wind velocity value of the surface velocity distribution diagram is at the place of 610 cm, Figure 8d shows that when the uniform distribution of the inflow velocity is 1.51 m/s and when the inflow average velocity is 5.275 m/s, the maximum wind velocity value of the surface velocity distribution diagram is at the place of 610 cm, Figure 8e shows that when the uniform distribution of the inflow velocity is 1.63 m/s and when the inflow average velocity is 4.46 m/s, the maximum wind velocity value of the surface velocity distribution diagram is at the place of 824 cm, Figure 8f shows that when the uniform distribution of the inflow velocity is 1.75 m/s and when the inflow average velocity is 4.84 m/s, the maximum wind velocity value of the surface velocity distribution diagram is at the place of 392 cm, Figure 8g shows that when the uniform distribution of the inflow velocity is 1.64 m/s and when the inflow average velocity is 4.95 m/s, the maximum wind velocity value of the surface velocity distribution diagram is at the place of 808 cm, Figure 8h shows that when the uniform distribution of the inflow velocity is 1.58 m/s and when the inflow average velocity is 4.57 m/s, the maximum wind velocity value of the surface velocity distribution diagram is at the place of 824 cm. It can be seen from Figure 8 that even if the inflow velocity distribution is not uniform, there will be a uniform phenomenon in the flow distribution of each chamber, and the peak value of the outlet side of the curved surface with high flow velocity will be on the high side, which is a common phenomenon. When the inflow velocity distribution is uniform, with the velocity surface distribution in Figure 8, the flow distribution in each chamber is more uneven compared with the time when the inflow velocity distribution is uniform, this is because the uneven phenomenon directly affected by the uneven distribution of air velocity leads to the uneven changes of wind velocity backflow and pressure, that is, uneven changes of backflow pressure cause the nonuniformity of flow, and the occurrence of the phenomenon of the flow with wind velocity pressure drop indicates that the uneven distribution of the inflow velocity causes the uneven distribution of wind velocity and flow in each chamber, and this uneven wind velocity phenomenon is more serious in the different air inlet configuration, which will also cause the difference in the heat taken away at each velocity outlet, this is consistent with the conclusion of Adamu [7]. Figure 8e, H shows the highest outlet velocity peak value, while the inflow air velocity surface peak value is in low distribution, it will change the average velocity of air inflow, and there will be a change in the reduction of heat exchange amount. When the average velocity is less than 1.6 m/s in the figure, the decrease of heat exchange amount increases with the increase of the average velocity, and the magnitude of increase is very big, this is because of the average velocity distribution, and when the average value of the inflow velocity distributed is bigger and bigger, the slope of the velocity surface distribution expression is bigger, namely, the bigger the average velocity is, the more uneven the head-on wind velocity distribution is, and the amplitude of attenuation is more and more gentle, this is because with the increase of the average velocity, the air wind resistance becomes smaller and smaller.

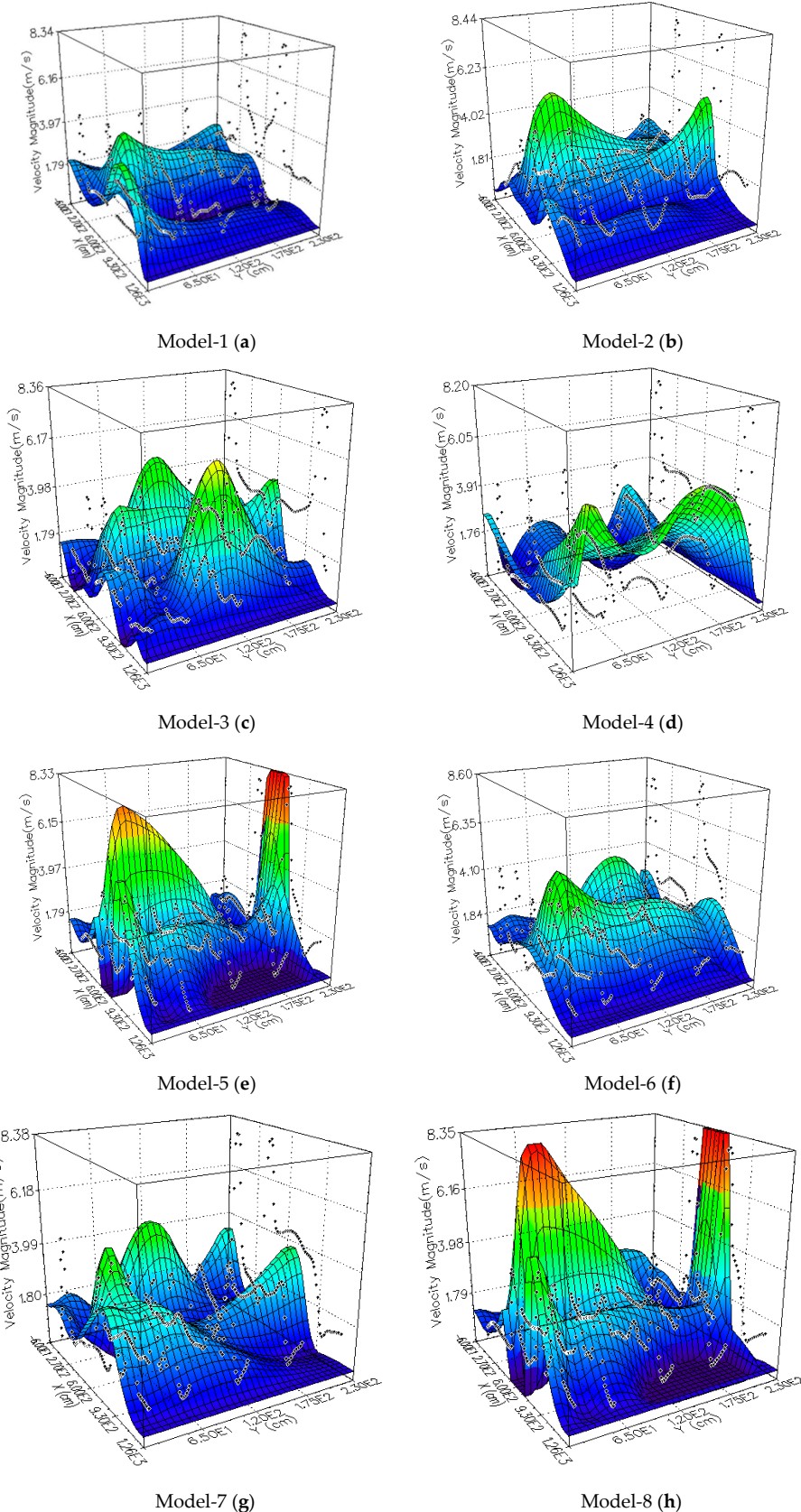

**Figure 8.** Velocity Magnitude Surface Plot (m/s); (**a**) Wind speed changes in container houses; (**b**) Wind speed changes in container houses; (**c**) Wind speed changes in container houses; (**d**) Wind speed changes in container houses; (**e**) Wind speed changes in container houses; (**f**) Wind speed changes in container houses; (**g**) Wind speed changes in container houses; (**h**) Wind speed changes in container houses.

*5.2. Analysis of Natural Ventilation Contour Plot of the Container House*

Figure 9 shows the air velocity as the profile of the result. Eddy behavior can be seen by operating the part of the inlet wind velocity distribution and from around the contour plot of the model of the indoor air velocity distribution with the reference wind velocity of 10 m/s, because the air layer is under natural ventilation, air viscosity and molecular attraction transfer to the air, the dense area in Figure 9c Model 3 is at the place of the contour plot with the maximum velocity. But this velocity increases the airflow in the direction of the air belt. Figure 9d as the opening and outlet of the container house, the distance configuration between the two is increased, but the opening of one more door is added between them, which significantly increases the wind velocity. Therefore, the potential wind velocity realized by the entrance of the more concentrated container house as shown in Figure 9 is air ventilation, and the indoor air velocity decreases with the increase of the window distance. Figure 9a shows the possibility of maximum natural ventilation in the middle of the air velocity distribution along the *X*-axis of the container house. To study the impact on the air, the indoor partition is here, with only the simulation results of the container house, as shown in Figure 9. The more uniform requirement for flow distribution of indoor air in both cases seems to be satisfied in part of most container houses. The establishment of the air velocity distribution along the *Z*-axis in the middle is compared in Figure 9. Figure 9 shows the case of the symmetrical parts of the eight models of the airflow pattern. The airflow patterns of all models have different velocity directions, and there is the shape of a large vortex at the place of 400 cm of the contour plot in Model-1 (Figure 9a), its position is in the center of the house, although not always the same. Similar phenomenon is found in Model-12 and Model-3, but the flow patterns between completely different cases are obtained.

The result of the high air exchange rate between the container house and the supplied air, and the mixed convection formed by the asymmetric open window from the configuration position of the windows and door in Figure 9c Mode 3 affect the accelerated air supply, which extends all over the entire indoor ventilation ultimately because of the action of inlet air pressure, therefore, the positions of opening air supply and exhaust outlet are relatively important. In addition, the effect of recirculating flow occurs due to the deceleration development on the right side near the door. Figure 9e,f,h show that the natural wind on the left has a relatively low flow rate on the area on the right side of the air supply and on the right side of the exhaust outlet. Therefore, because of the influence of the flow rate of the air supply, the contour plot shows that the rest are almost the same.

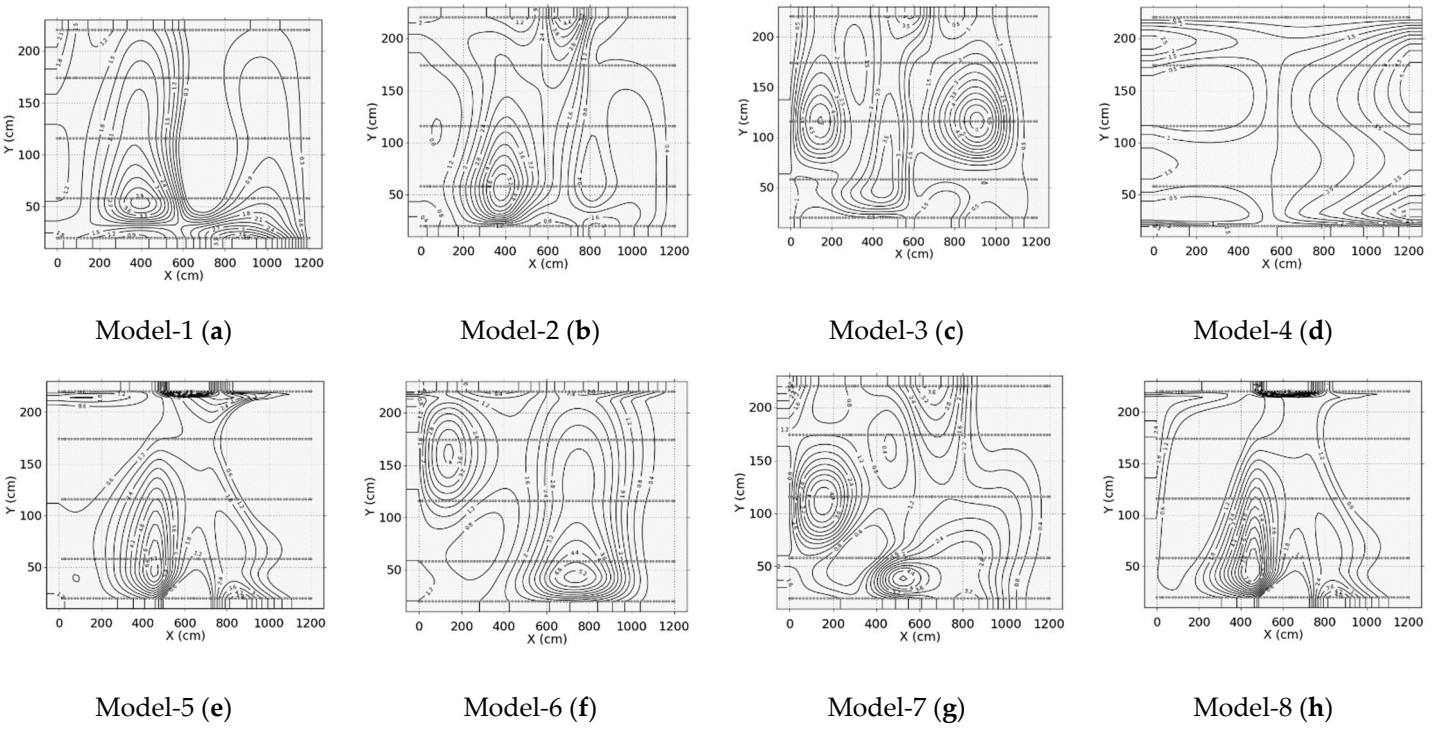

Model-1 (**a**)          Model-2 (**b**)          Model-3 (**c**)          Model-4 (**d**)

Model-5 (**e**)          Model-6 (**f**)          Model-7 (**g**)          Model-8 (**h**)

**Figure 9.** Velocity Magnitude (m/s) Contour Plot; (**a**) Wind speed changes in container houses; (**b**) Wind speed changes in container houses; (**c**) Wind speed changes in container houses; (**d**) Wind speed changes in container houses; (**e**) Wind speed changes in container houses; (**f**) Wind speed changes in container houses; (**g**) Wind speed changes in container houses; (**h**) Wind speed changes in container houses.

*5.3. Analysis of Solar Radiation Thermal Field in the Container House*

Four different temperature fields are mainly analyzed, namely 25°, 30°, 35° and 40°, as mentioned earlier, for the purposes of CFD simulating and predicting the air flow patterns in the container house. The main concern is the air flow in the container house to explore the dynamic state of wind flow under natural ventilation in order to obtain more information for the analysis of wind velocity affecting the thermal field. Because of the complexity of the flow field of the container house, the position of the entrance seems to affect the appropriate approach. Fresh air is provided to the container house, and is located in the front through three openings. The radiant heat is received from the surface of the container house, CFD boundary conditions are especially sensitive wind velocity flow, when the air flow and unreasonable pressure distribution combination model are found, zero pressure boundary conditions are defined as the computational domain of the side boundary and the upper boundary.

The optimal convective heat flux is the maximum value that the temperature difference between the air supplied and the air change rate reaches in the middle position 600 cm, and the air supply and discharge position is very important, meanwhile it affects the determination of radiation heat distribution. In this case, at the center line air supply position, the optimal placement position for exhaust is in Model 3 and Model 4. The reduction of convective heat flux makes the discharge be close to 0–200 cm. Air temperature reduces, it may changes in a curved-surface manner as the surrounding air temperature may increase, the surrounding surface temperature change is not conducive to natural ventilation, so as to reduce the temperature of the indoor air, which will be a negative impact on the temperature of the indoor air in ventilation. The following is a comparative difference diagram between eight models at different temperatures.

The container house Model-1 Figure 10 shows that the maximum indoor temperature is 50.7 °C due to the effect of solar radiation and ambient temperature 25 °C, while the temperature at the middle position 600 cm is 36 °C. The effect of natural ventilation can be clearly seen, the convection phenomenon in the container house can reduce the maximum temperature of the container house under the solar radiation effect by about 15 °C, while the lowest indoor temperature can be maintained at about 27 °C. The indoor ambient temperature rising to 30 °C will keep heat the surface of the container house due to solar radiation, plus the surrounding ambient temperature factor, so that the indoor temperature maintains at a stable state Figure 10b. However, when the ambient temperature reaches 35 °C, the temperature field change will increase and be more intense in the central part, therefore, the mass flow rate of the air outlet is more closely related, the temperature gradient change is smaller, thus forming a three-dimensional curved surface distribution diagram of uneven temperature field distribution direction (Figure 10c,d). Because the air outlet is located in the middle of the space and the inlet is at the asymmetric position, it increases the chance that the cold air in this area will take away the hot air from the inlet velocity. However, because of the configuration of the air inlet, the cold air at the air outlet in the central area can have full cyclic action in the space, therefore, when the effect of thermal energy (1000 w/m$^2$) emitted by the solar radiation of the container house exerts an influence, it is located in the central area, therefore, the mass flow rate of the air outlet is more closely related, the temperature gradient change is smaller, thus forming an uneven temperature field distribution.

Figure 11a of the container house Model-2 shows that the maximum indoor temperature is 44.9 °C due to the effect of solar radiation and ambient temperature of 25 °C. The temperature at the low temperature position of 613cm is 27 °C, so that the maximum temperature of the container house under the solar radiation effect can be reduced by about 18 °C. When the indoor ambient temperature rises to 30 °C, the solar radiation continues to heat the surface of the container house, plus the ambient temperature factor, the maximum indoor temperature can reach 51 °C Figure 11b. However, when the ambient temperature reaches 35 °C and 40 °C, the change of the temperature field is relatively stable, while from 100 cm to the part located in the central area, there is a small temperature gradient change, 800–1000 cm forms an uneven temperature field distribution as shown in Figure 11c,d. Since the air inlet and outlet configurations are both located in the middle of the space, the cold air in this area is too concentrated so as to affect the flow rate to take away the hot air on both sides of the space, so the mass

flow rate at the air outlet is more related, and there is a relatively small temperature gradient change, which forms the condition to influence two lateral uneven temperature field distribution. When the ambient temperature reaches 40 °C, the indoor ambient temperature rises to 63 °C.

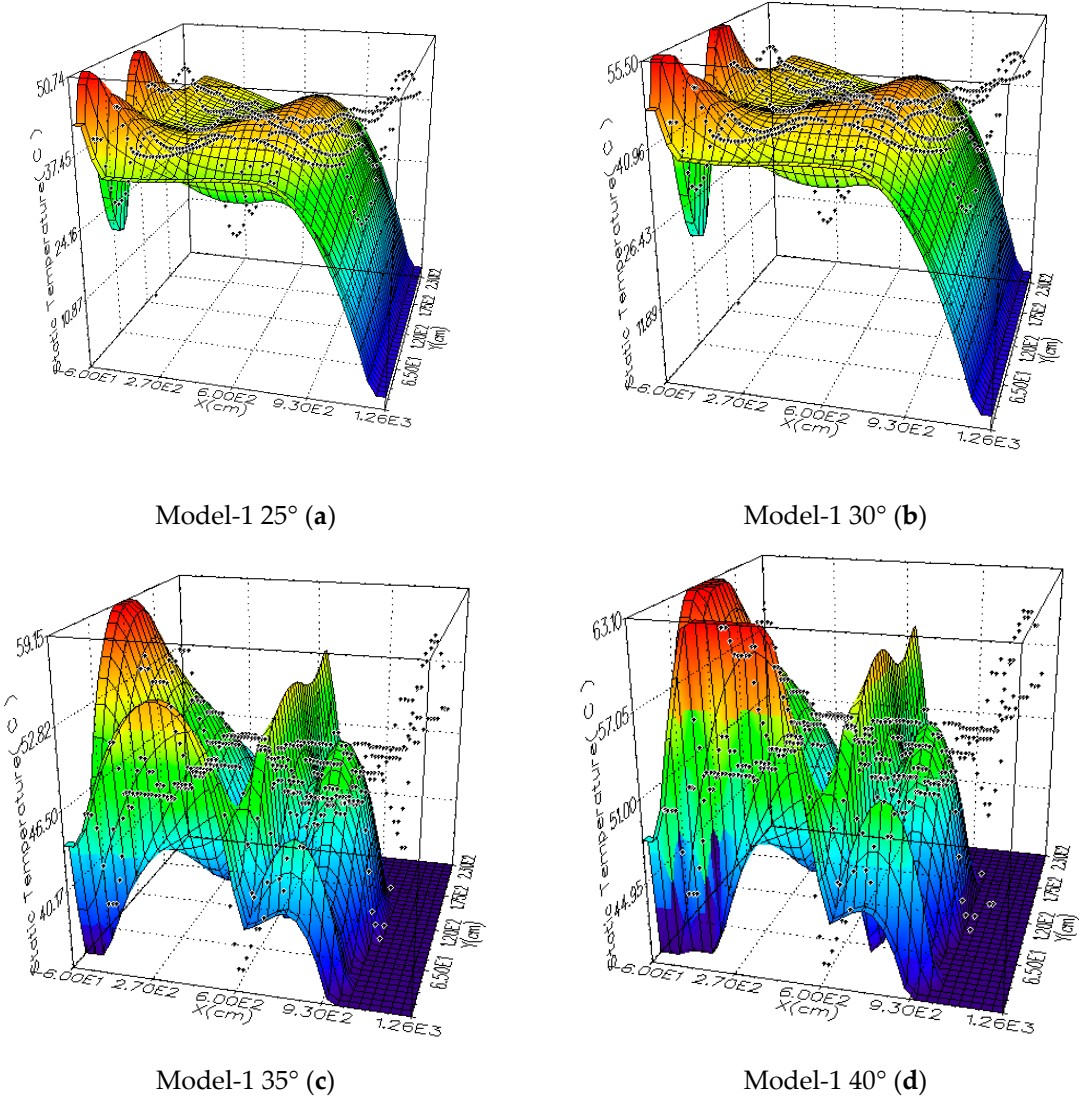

Model-1 25° (**a**)  Model-1 30° (**b**)

Model-1 35° (**c**)  Model-1 40° (**d**)

**Figure 10.** Model-1 Static Temperature Surface Plot (°C); (**a**) temperature change; (**b**) temperature change; (**c**) temperature change; (**d**) temperature change.

Figure 12 of the container house Model-3 shows that the maximum indoor temperature is 43.9 °C due to the effect of solar radiation and ambient temperature of 25 °C; the low temperature is 26 °C at the window position of 584 cm, the temperature difference is about 18 °C, when the indoor ambient temperature rises to 30 °C, the solar radiation continues to heat the surface of the container house, plus the ambient temperature factor, the maximum indoor temperature can reach 48.9 °C as shown in Figure 12b. When the ambient temperature reaches 35 °C, the change of temperature field is similar to that at 30 °C, but the maximum indoor temperature reaches 52.8 °C, the highest temperature is located in the area of 900 cm, but when the environment temperature reaches 40 °C, uniform temperature field distribution is as shown in Figure 12d, and indoor temperature falls below 60 °C, the temperature is lower than other temperature field. The model is located in the middle of the space due to the air inlet configuration, the asymmetric fluid flows on both sides of the container house at the outlet, so the cold air can effectively take away the radiant heat in the space. The mass flow rate of some air inlet in the

central area has bigger high pressure wind velocity relationship, which results in a relatively uniform temperature field distribution.

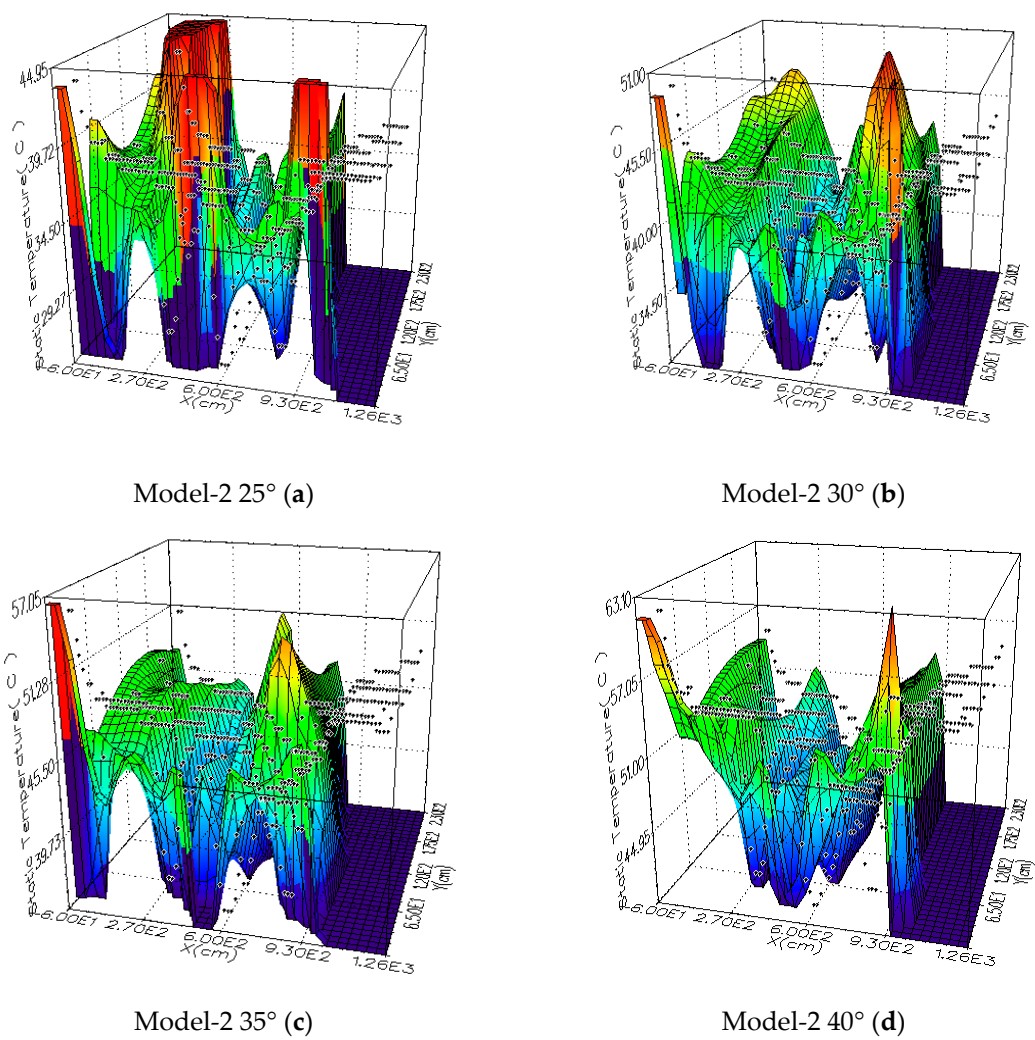

<center>Model-2 25° (**a**)　　　　　　　　　　　Model-2 30° (**b**)</center>

<center>Model-2 35° (**c**)　　　　　　　　　　　Model-2 40° (**d**)</center>

**Figure 11.** Model-2 Static Temperature Surface Plot (°C); (**a**) temperature change; (**b**) temperature change; (**c**) temperature change; (**d**) temperature change.

In Figure 13d of the container house Model-4, the model is located in the position between the two sides of the space due to the air inlet configuration. The inlet and outlet run from opposite directions, which leads to the wind velocity moving directly from the inlet to the outlet, so the cold air is unable to effectively take away excess radiant heat, forming an uneven temperature field distribution in the middle of the container house. The maximum indoor temperature is 58.6 °C due to the effect of solar radiation and ambient temperature of 40 °C, the indoor temperature is lower than other models, but the intermediate temperature is in the high environment, which affect the area where people often carry out activities.

Figure 14 of the container house Model-5 shows that the change in the air inlet position of the door results in that the maximum indoor temperature has reached 52 °C as shown in Figure 14a due to the effect of solar radiation and ambient temperature of 25 °C, when compared with the positions of the aforementioned four doors in the middle. However, when the ambient temperature reaches 40 °C, the temperature of the temperature field changes to 67.47 °C, the temperature reaches the highest in the area of 900 cm, and there is also a larger temperature gradient change, forming an uneven temperature field distribution as shown in Figure 14d. Since the position of the most important door

inlet configuration changes to the right side of the space, the cold air on the left side of the area can not effectively take away solar radiation heat in the space. Therefore, when the ambient temperature reaches 40 °C, the indoor ambient temperature will rise to 67.7 °C.

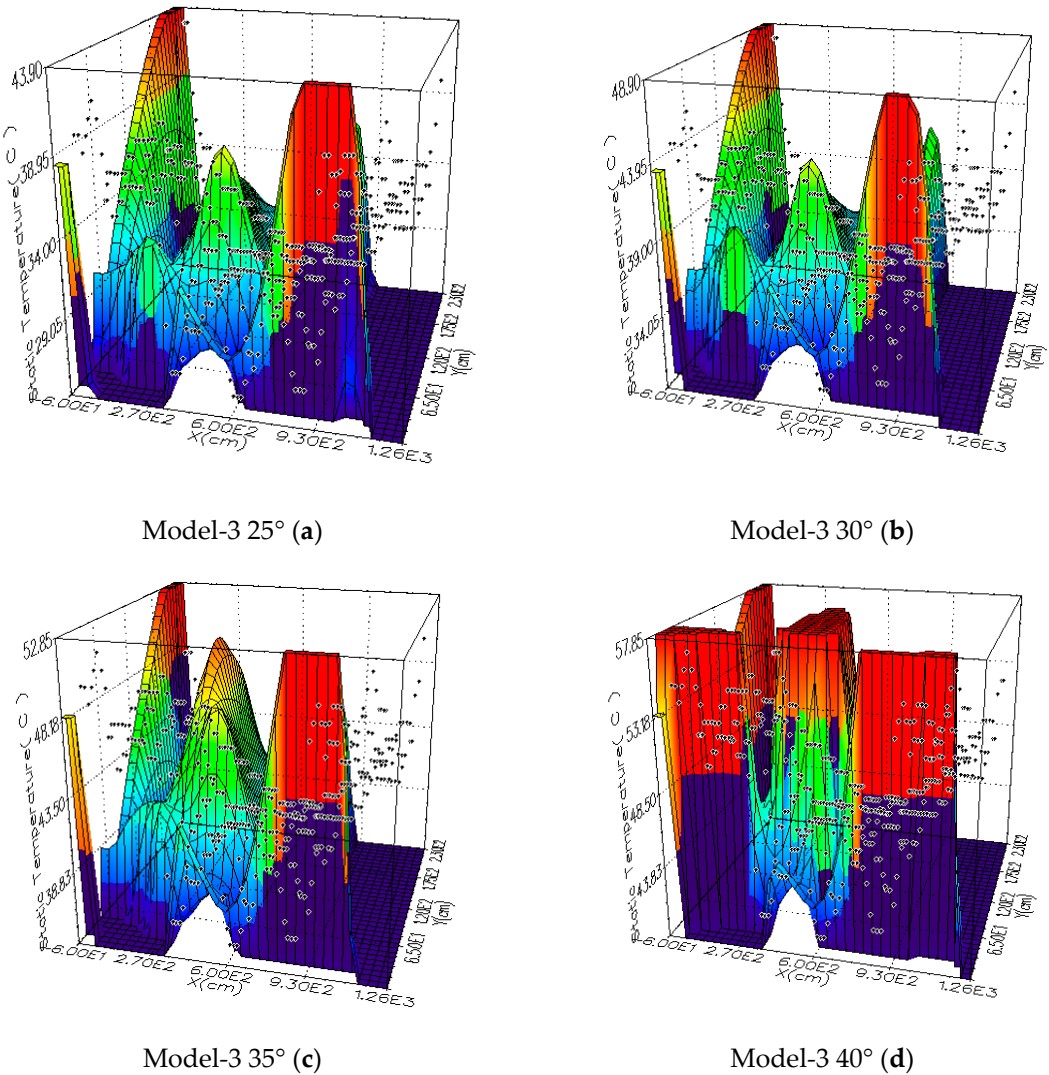

Model-3 25° (**a**)                                    Model-3 30° (**b**)

Model-3 35° (**c**)                                    Model-3 40° (**d**)

**Figure 12.** Model-3 Static Temperature Surface Plot (°C); (**a**) temperature change; (**b**) temperature change; (**c**) temperature change; (**d**) temperature change.

Figure 15 of the container house Model-6 shows that the change in the air inlet position of window 4 affects the change of the whole temperature field, which results in that the maximum indoor temperature has reached 51.25 °C as shown in Figure 15a due to the effect of solar radiation and ambient temperature of 25 °C. When the ambient temperature reaches 35 °C and the temperature field changes, the temperature field discharges air by shifting from the left side to the right side along window 4. Figure 15c clearly shows that the temperature gradient belt shifts from the position of 270 cm with temperature of 36 °C to the position of 870 cm with temperature of 37 °C.

Figure 16 of the container house Model-7 shows that the maximum indoor temperature is 53.4 °C due to the effect of solar radiation and ambient temperature of 25 °C. With the change in the air inlet position of window 2, asymmetry of the window and and effect of the wind on the cross-flow velocity, the effect of natural ventilation can be clearly seen. The convective phenomenon in the container house makes the maximum temperature reduce by 4 °C when the container house under the effect of solar radiation and the ambient temperature of 40 °C in Figure 14d of Model-6 are compared. However,

the effect of the window opening size and Models 1–4 can not effectively take away radiant heat from the indoor air flow in the central area, so the air outlet and mass flow rate are more related, and there is a smaller temperature gradient change, forming more uniform temperature field distribution in the middle.

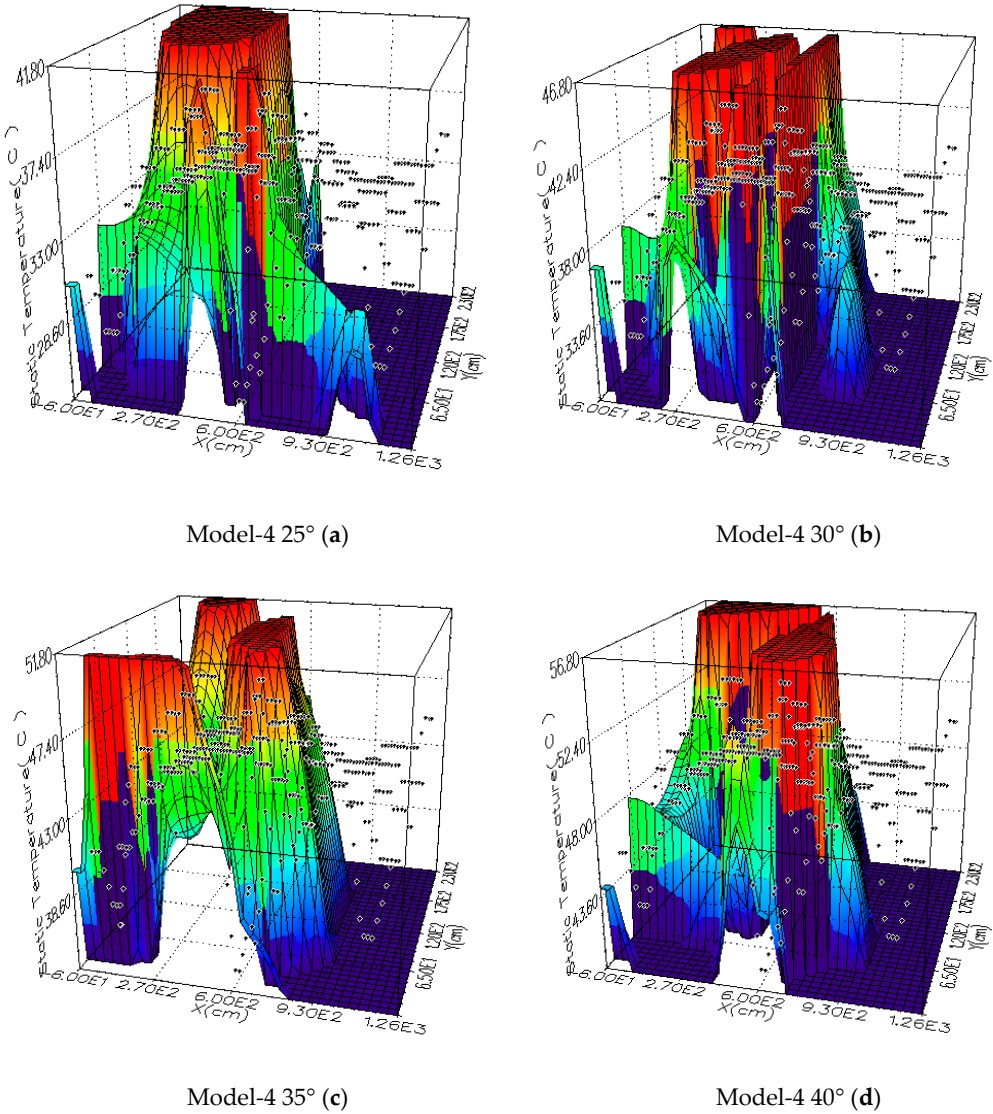

Model-4 25° (**a**)  Model-4 30° (**b**)

Model-4 35° (**c**)  Model-4 40° (**d**)

**Figure 13.** Model-4 Static Temperature Surface Plot (°C); (**a**) temperature change; (**b**) temperature change; (**c**) temperature change; (**d**) temperature change.

Figure 17 of the container house Model-8 shows that as the air outlet configuration is located on the right side of the space, resulting in the problem of excessive concentration of cold air in the entry area. When the ambient temperature is 40 °C, the highest indoor temperature in the eight models reaches as high as 78.04 °C, so the air inlet configuration influence factor is very high, the cold air can fully circulate in the space, the air inlet and outlet of natural ventilation are very important influence factors, and the asymmetric opening produces cross air flow, which can form a uniform distribution of temperature field. And the eddy current also has considerable effect on the entire indoor air flow of the container house, so the effect of natural ventilation is obvious.

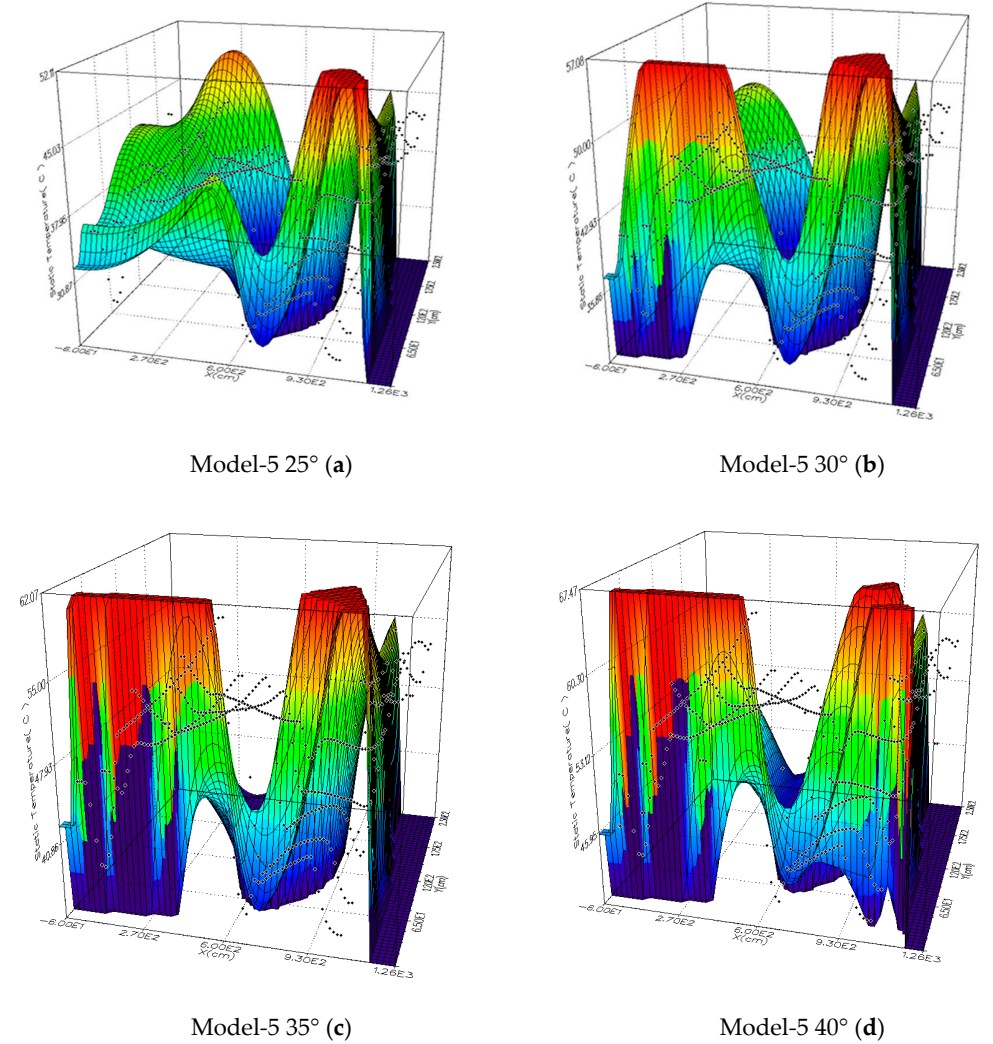

Model-5 25° (**a**)

Model-5 30° (**b**)

Model-5 35° (**c**)

Model-5 40° (**d**)

**Figure 14.** Model-5 Static Temperature Surface Plot (°C); (**a**) temperature change; (**b**) temperature change; (**c**) temperature change; (**d**) temperature change.

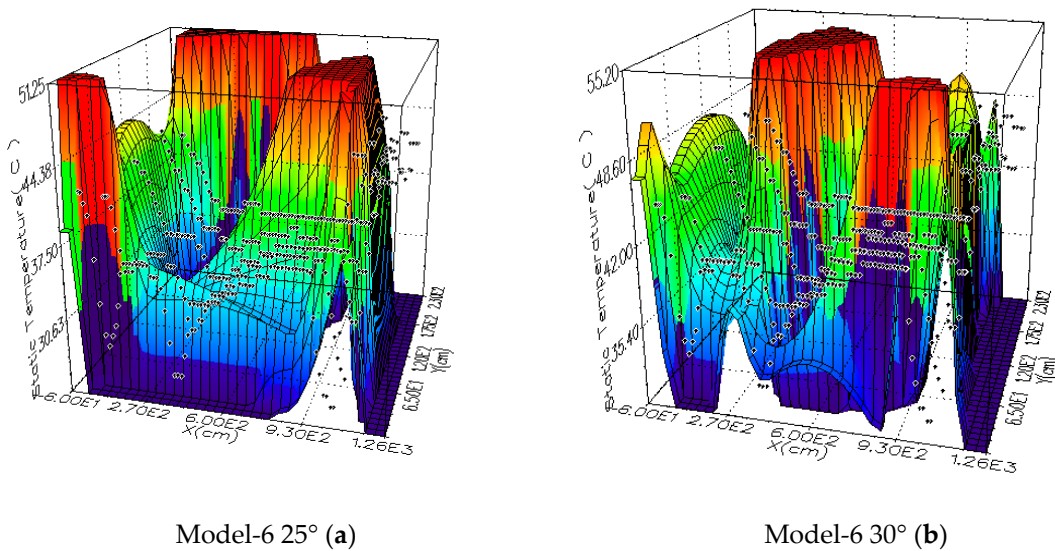

Model-6 25° (**a**)

Model-6 30° (**b**)

**Figure 15.** *Cont*.

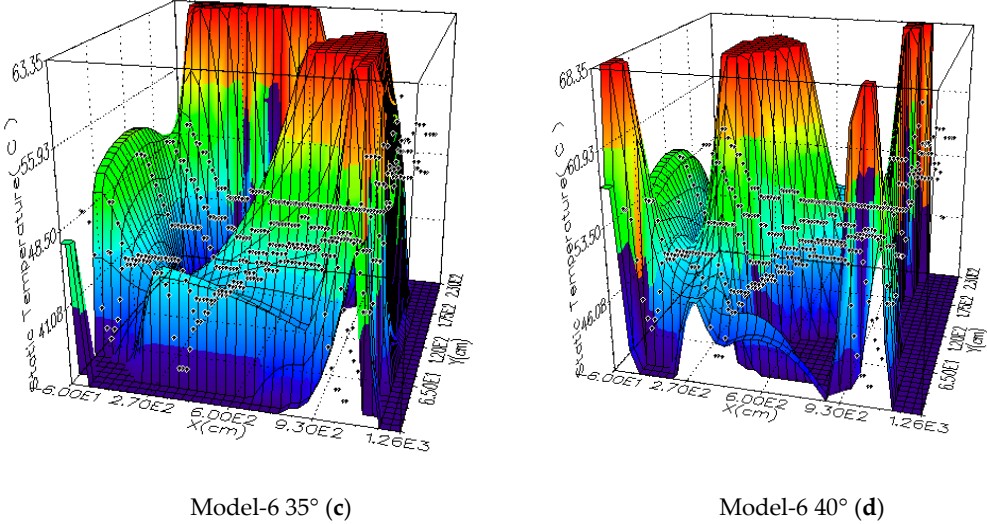

Model-6 35° (**c**)                Model-6 40° (**d**)

**Figure 15.** Model-6 Static Temperature Surface Plot (°C); (**a**) temperature change; (**b**) temperature change; (**c**) temperature change; (**d**) temperature change.

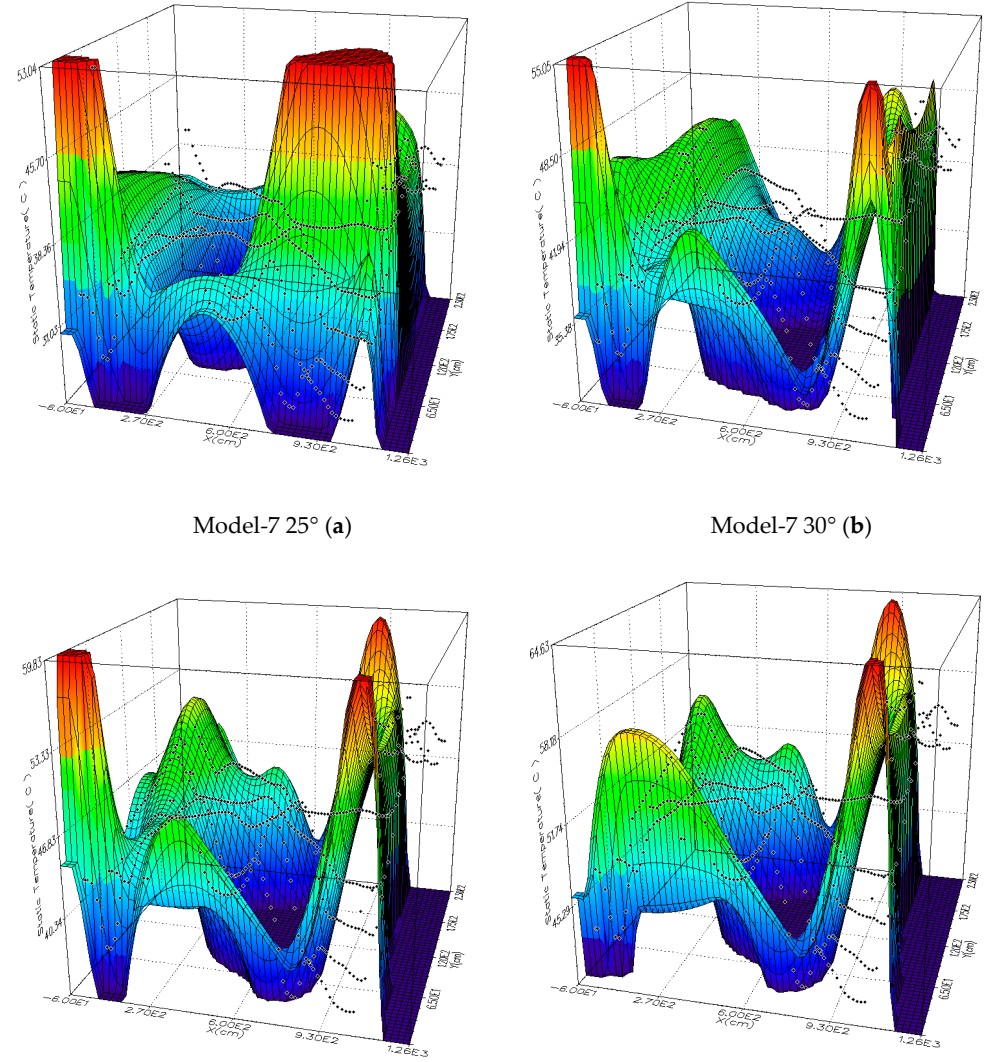

Model-7 25° (**a**)                Model-7 30° (**b**)

**Figure 16.** Model-7 Static Temperature Surface Plot (°C); (**a**) temperature change; (**b**) temperature change; (**c**) temperature change; (**d**) temperature change.

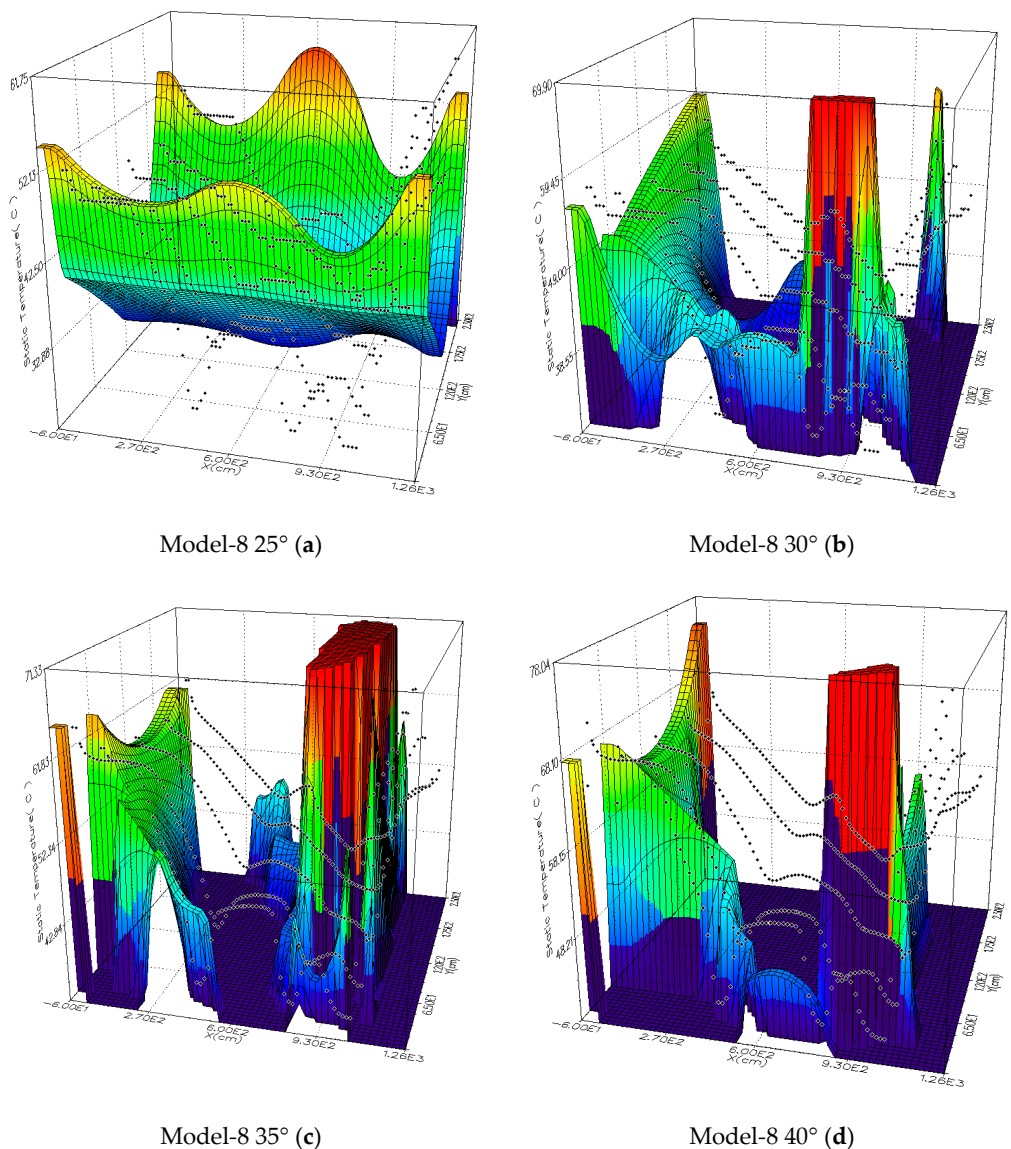

Model-8 25° (**a**)  Model-8 30° (**b**)

Model-8 35° (**c**)  Model-8 40° (**d**)

**Figure 17.** Model-8 Static Temperature Surface Plot (°C); (**a**) temperature change; (**b**) temperature change; (**c**) temperature change; (**d**) temperature change.

### 5.4. Comparative Curve Analysis of Solar Radiant Heat in the Container House

These eight sets of CFD models seem to provide reasonable flow patterns and container house temperature distribution characteristics. The values of velocity and temperature range are different from each other. In order to fit the comparison in the airflow rate in the container house and the combined model between the two speed and temperature modes. These conditions determine the flow pattern and the temperature distribution in the space. The optimal thermal convection flux is the maximum temperature difference between the solar radiant heat and the supply air and the rate of air.change reaching its maximum value. On the other hand, as shown in Temperature Comparison Curve Chart in Figure 18, in this case, natural ventilation takes away the heat in the middle, and air exhaust can be anywhere. For this question, Figure 18 shows the results, so the temperature curves close to 200 cm on the right and 1000 cm on the left are ignored.

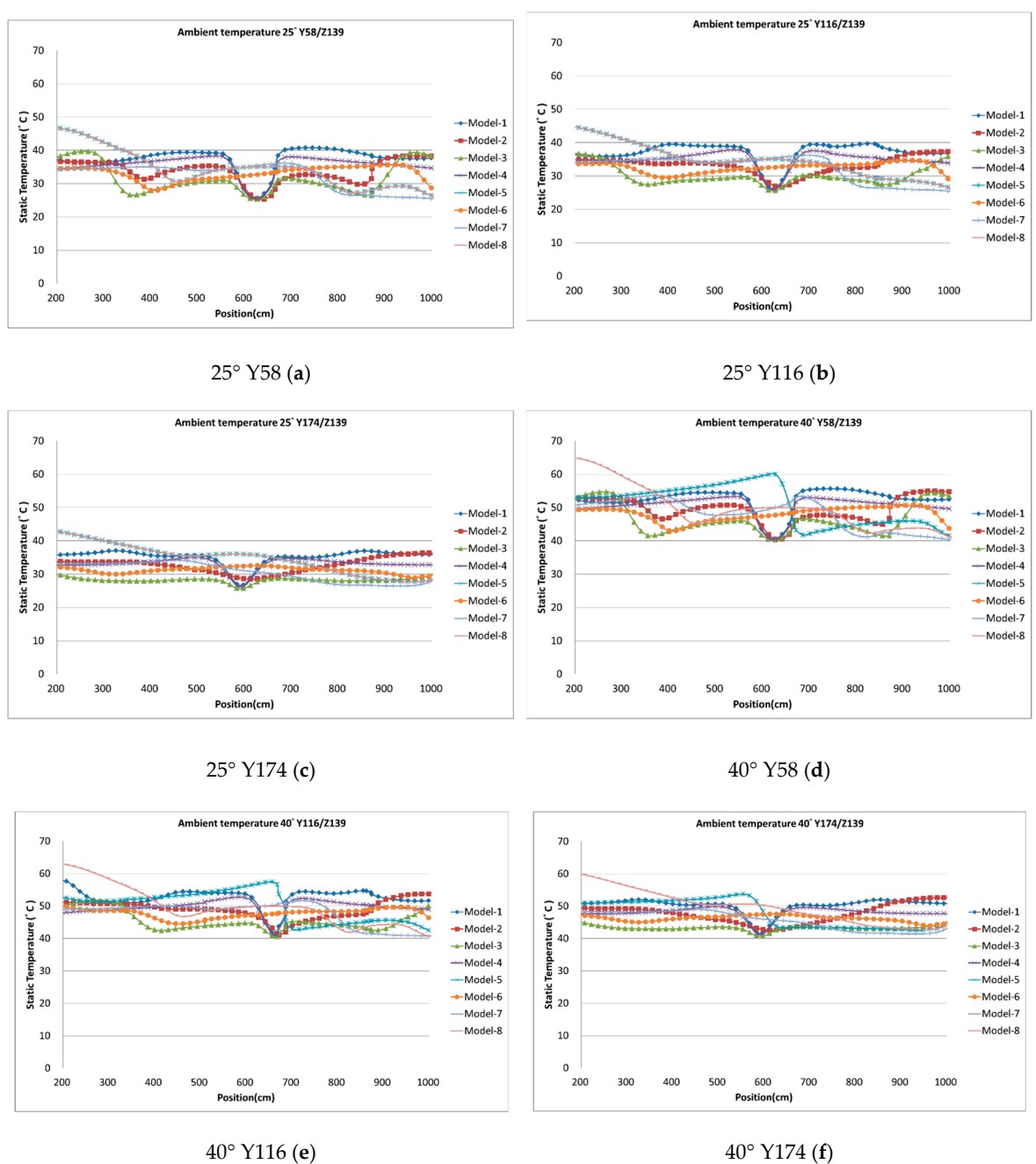

**Figure 18.** Temperature Comparison Curve Chart; (**a**) Wind speed changes in container houses; (**b**) Wind speed changes in container houses; (**c**) Wind speed changes in container houses; (**d**) Wind speed changes in container houses; (**e**) Wind speed changes in container houses; (**f**) Wind speed changes in container houses.

*5.5. Analysis of Temperature Change and Wind Velocity Change in the Container House*

To understand whether the wind velocity has an effect under the temperature change, the total exhaust air volume of the air outlet is measured, and the measurement of the direction of the air flow changes from the same wind velocity and temperature and from 25 to 40 °C. During the measurement, the indoor air flow velocity is also relatively increased due to changes in temperature, Model 5 with temperature of 40 °C increases wind velocity by 7.02 m/s as shown in Figure 19 The items are described in Table 6 below:

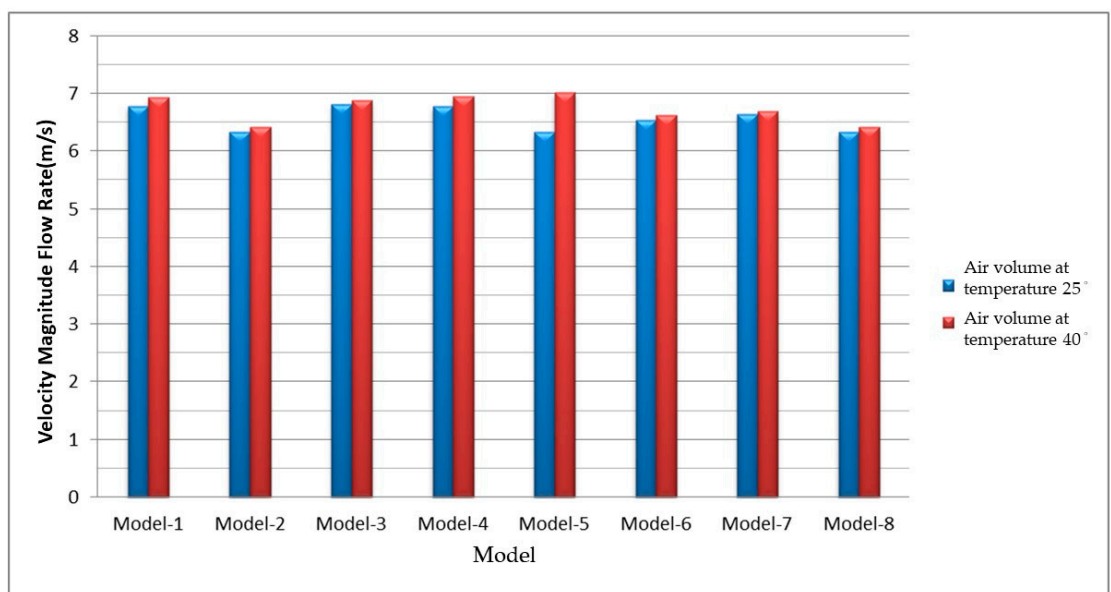

**Figure 19.** Velocity Magnitude Flow Rate(m/s).

**Table 6.** Temperature change flow chart.

|  | Model-1 | Model-2 | Model-3 | Model-4 | Model-5 | Model-6 | Model-7 | Model-8 |
|---|---|---|---|---|---|---|---|---|
| Air volume at 25 °C | 6.79 | 6.34 | 6.82 | 6.78 | 6.33 | 6.54 | 6.64 | 6.33 |
| Air volume at 40 °C | 6.93 | 6.43 | 6.89 | 6.95 | 7.02 | 6.63 | 6.69 | 6.42 |

Natural ventilation in the Y direction with an indoor air velocity distribution of 10 m/s generally meets the requirements of thermal comfort for all eight models, especially for model 3, which has a relatively high average natural ventilation, air flows from the inlet to the indoor air. The requirements of thermal comfort of all models in velocity distribution are met. The average value of air velocity can be easily seen, assuming that the configured inlet may increase the indoor air velocity and extend the thermal comfort time under the influence of solar radiant heat. Formulated air velocity and radiant heat and validated CFD are the main tools of the container house. The indoor air velocity is used to calculate the air flow effect of these velocities under 8 different model conditions so as to improve the operating temperature change in the container house, while the relatively high indoor air velocity in the container house can enhance the thermal comfort. However, this natural ventilation method can not provide the thermal comfort of the container house under high ambient temperature. In real life, more container houses will be connected to the parallel mode of concentration, so the air velocity in the container house will almost certainly decrease. Under the future study on ventilation of container houses, a centralized container house study will be conducted to conduct the optimal configuration and take into account the optimal ambient temperature time. Finally, natural ventilation is the most efficient ventilation system with 100% energy saving. If it can be applied, it has the most environment-friendly effect on earth.

## 6. Conclusions

This study analyzes the natural ventilation of container houses. Container houses are temporary homes in the event of disasters, which can provide people with a simple living environment in a naturally ventilated environment. Further research is carried out to change the door and window dimensions by adding some air inlet and outlet sections. The effects of air movement and solar radiant heat in the container house at different air inlet configuration positions are discussed. The purpose of this study is to investigate whether the position of the opening of the container house can improve natural ventilation. The methods adopted are CFD simulation of eight basic models and solar radiation

under four ambient temperatures, and are compared with relevant literature for verification. Parameter studies include air change rate, wind velocity, wind distribution and its homogeneity. The results of this study are shown below:

I.   The experimental results, which are quite close to the CFD prediction, confirm that the eight models form a good prediction of the simple configuration of the container house and indoor airflow.

II.  In terms of the ventilation and indoor airflow patterns under different configurations of windows and doors, the effect of model 3 is the best, and this effect is very significant. In terms of indoor wind velocity, under the asymmetry of wind velocity of asymmetric windows, and under the comparison with the thermal effects taken away by ventilation and airflow, the gaps of air quality and thermal comfort are shown by the analysis model. When the volumes of the container houses are the same and the velocity is consistent with the solar radiant heat, there is a 10% gap in the maximum flow velocity between the models.

III. Temporary use of container houses plays an important role in this study. When the ambient temperature increases, the airflow velocity of natural ventilation increases, and the effect of the window configuration position is seen, indoor airflow depends on various parameters and characteristics of the incoming air flow, therefore, when the indoor airflow is analyzed, many design changes are also analyzed at the same time, such as the potential of natural ventilation and cooling temperature field.

IV.  It is found from the study that the wind velocity at the window outlet can also be increased by the natural ventilation of the door at the entrance, which can increase the natural ventilation velocity of the outlet window, and the airflow generated by the door ventilation opening in the house can take away a large amount of radiant heat air movement.

V.   The study results show that the configuration position of the inlet and outlet sections of the container house can improve the efficiency of natural ventilation, so the results show that when the temperature is 25 degrees, the indoor thermal environment temperature at the central position of model 3 decreases the benefit by 35% that of model 8, which can illustrate that 300 cm–900 cm is the optimal position for thermal comfort in the activity area of the container house.

VI.  Among various types of container houses, model 3 can achieve the optimal effect, its sunshine environment is 1041 W/m2, while the living environment temperature in container houses is within 45 °C, which is acceptable to the human body.

This paper is about the study on the effects of solar radiation on different container house models, and the solar radiant heat carried away by the airflow generated by natural wind in the house is calculated. Based on fluid dynamics, the system characteristics with complexity and natural ventilation are analyzed to establish a complete understanding of airflow through multiple windows. This study has found that in various cases, the natural ventilation effect of the openings of the asymmetric window with better performance in various aspects is the best. This study can help solve the housing problem most effectively when disasters occur, and can also let relevant design units understand the prediction effect of indoor airflow in container houses.

Sustainability is a design method that can be used to save energy, water resources, and building materials to avoid pollution to the interior and exterior of the building. It is called green design to create an ideal ecological environment.

**Author Contributions:** The author contributed to the paper. H.-H.L. Collects and organizes data and acts as the corresponding author, J.-H.C. and the authors propose methods. All authors have read and agreed to the published version of the manuscript.

**Funding:** This work was supported by the Ministry of Science and Technology of the Republic of China under grant MOST 109-2511-H-992-001.

**Conflicts of Interest:** The authors declare no conflict of interest.

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
