# Peer review of "A Study of the Simulation and Analysis of the Flow Field of Natural Convection for a Container House"

_sustainability, doi:10.3390/su12239845_

Round 1

Reviewer 1 Report

Overall idea is really nice, however, 

need major revision before any decision. 

Authors must added all comments answers in revised version to strengthen their article.

Comment file is attached as pdf.

Author Response

The revised manuscript of the above paper entitled “A Study of the Simulation andAnalysis of the Flow Field of NaturalConvection for a Container House” has been uploaded along with a separate list of point-by-point response to the comments raised by the reviewers. We would like to take this opportunity to express our sincere thanks to the reviewers who identified areas of our manuscript that needed corrections or modification. We would also like to thank you for allowing us to resubmit a revised copy of the manuscript.

Reviewer 2 Report

Firstly, the manuscript is not well prepared with a lot of grammar errors. The presenting is really confusing at some places. I would recommend the authors to re-write and deep proofreading before submission. Secondly, the reviewer does not think the research topic has a high value to draw a high interest for the readers. Most of the conclusions are general knowledge, which obviously stands, for example the effect of window size and position. In addition, how the study of natural ventilation could help to improve the design of a container house should be discussed.

  1. Grammar errors: line #16, line #35, line 131, line 141, line 200, etc…
  2. Could the author break up the sentence from line 40 to line 44? The presenting is really not clear.

Author Response

(The authors gave the same response as above.)

Round 2

Reviewer 1 Report

Revision is fine now.

Author Response

(The authors gave the same response as above.)

Reviewer 2 Report

There is no significant improvement in the manuscript revision.

Author Response

已上傳上述論文的修訂稿,標題為“集裝箱房屋自然對流流場的模擬與分析研究”,以及針對審稿人提出的意見逐點回應的單獨清單。我們想藉此機會向審稿人表示衷心的感謝,他們確定了我們手稿中需要更正或修改的區域。我們還要感謝您允許我們重新提交手稿的修訂本。
